# On Sketching for Gaussian Process Regression with New Statistical Guarantees

**Jayesh Malaviya***
*Department of Computer Science & Engineering*
*Indian Institute of Technology, Gandhinagar*

*malaviya_jayesh@iitgn.ac.in*

**Rachit Chhaya**
*Department of Computer Science & Engineering*
*DA-IICT, Gandhinagar*

*rachit_chhaya@daiict.ac.in*

**Anirban Dasgupta**
*Department of Computer Science & Engineering*
*Indian Institute of Technology, Gandhinagar*

*anirbandg@iitgn.ac.in*

**Supratim Shit**
*Department of Computer Science & Engineering*
*IIIT, Delhi*

*supratim@iiitd.ac.in*

**Reviewed on OpenReview:** *https://openreview.net/forum?id=NmwrhyuVEu*

## Abstract

The cubic computational complexity of Gaussian Process Regression (GPR) with respect to the number of data points is a major bottleneck to its scalability. While various approaches have been proposed to address this, few come with provable guarantees. Inspired by the success of ridge leverage score based sampling in scaling kernel ridge regression El Alaoui & Mahoney (2015), we propose a sketch-based approximation for GPR using ridge leverage scores. We provide theoretical guarantees on the approximation of the predictive mean, predictive variance, and negative log-marginal likelihood in this setting. To the best of our knowledge, these are the first theoretical guarantees for approximating the predictive variance and negative log-marginal likelihood of GPR using ridge leverage score sampling. We further show that a carefully constructed sketch of the kernel matrix preserves key statistical properties of the full GPR model with high probability. Our theoretical results are supported by empirical evaluations on real-world datasets, demonstrating strong trade-offs between accuracy and efficiency.

## 1 Introduction

Gaussian Process Regression (GPR) is a fundamental method in probabilistic machine learning, offering a principled non-parametric approach to modeling distributions over functions Rasmussen & Williams (2005). Its strength lies in its ability to provide calibrated uncertainty estimates, which are critical in applications such as Bayesian optimization Xu et al. (2024), active learning Kapoor et al. (2007); Schreiter et al. (2015); Tebbe et al. (2024), and reinforcement learning Bıyık et al. (2020).

**Gaussian Process Regression.** Given a training dataset $\mathcal{D} = \{(x_i, y_i)\}_{i=1}^n$, where $x_i \in \mathbb{R}^d$ and $y_i \in \mathbb{R}$, we assume the outputs are generated from a latent function $f \sim \mathcal{GP}(m(\cdot), k_\theta(\cdot, \cdot))$, corrupted by Gaussian noise, i.e.,

$$y_i = f(x_i) + \xi_i, \quad \xi_i \sim \mathcal{N}(0, \sigma_\xi^2).$$

---

* Corresponding author.

Here, $m(\cdot)$ is the prior mean function (often taken as constant), and $k_\theta(\cdot, \cdot)$ is a positive-definite kernel function parameterized by hyperparameters $\theta$. A commonly used choice for the kernel function is the Radial Basis Function (RBF) kernel, defined as,

$$k_\theta(x, x') = \sigma_f^2 \exp\left(-\frac{\|x - x'\|^2}{2\ell^2}\right) \tag{1}$$

where $\sigma_f^2$ controls the variance and $\ell$ is the lengthscale hyperparameter.

The prior over latent function values $\mathbf{f} = [f(x_1), \ldots, f(x_n)]^\top$ is multivariate Gaussian,

$$\mathbf{f} \sim \mathcal{N}(m(X), K)$$

where $K \in \mathbb{R}^{n \times n}$ is the kernel matrix with entries $K_{ij} = k_\theta(x_i, x_j)$, and $m(X)$ is the vector of prior means evaluated at training inputs.

Under this model, the noisy observations $\mathbf{y} \in \mathbb{R}^n$ are distributed as

$$\mathbf{y} \sim \mathcal{N}(m(X), K + \sigma_\xi^2 I)$$

**Prediction at Test Time.** For a new test point $x_*$, the predictive distribution of $y_*$ conditioned on the training data is also Gaussian,

$$y_* \mid x_*, X, \mathbf{y} \sim \mathcal{N}(\mu(x_*), \text{Var}(x_*))$$

where the predictive mean and variance are respectively given as,

$$\mu(x_*) = k_*^\top (K + \sigma_\xi^2 I)^{-1} \mathbf{y} \tag{2}$$

$$\text{Var}(x_*) = k(x_*, x_*) - k_*^\top (K + \sigma_\xi^2 I)^{-1} k_* \tag{3}$$

with $k_* = [k_\theta(x_*, x_1), \ldots, k_\theta(x_*, x_n)]^\top$

**Learning via Marginal Likelihood.** Given the data, the task in GPR is to learn the kernel hyperparameters $\theta$ and noise variance $\sigma_\xi^2$. This is typically done by maximizing the log marginal likelihood of the observed outputs,

$$\log p(\mathbf{y} \mid X, \theta) = -\frac{1}{2}\mathbf{y}^\top (K + \sigma_\xi^2 I)^{-1}\mathbf{y} - \frac{1}{2}\log\det(K + \sigma_\xi^2 I) - \frac{n}{2}\log 2\pi \tag{4}$$

This objective balances data fit (first term), model complexity (second term), and normalization.

**Computational Challenges.** The exact computation of equation 2 to equation 4 requires $\mathcal{O}(n^3)$ time and $\mathcal{O}(n^2)$ memory due to the inversion and determinant of the full kernel matrix Rasmussen & Williams (2005). This limits the applicability of standard GPR to small or moderate-sized datasets. In this work, we address this scalability bottleneck through a sketching-based approximation using ridge leverage scores.

To handle this problem, a rich line of work has focused on approximating the kernel matrix using techniques such as inducing points Snelson & Ghahramani (2006), Nyström methods Williams & Seeger (2001), and randomized sketching El Alaoui & Mahoney (2015); Pilanci & Wainwright (2017). Among these, sketching methods stand out for their ability to compress large kernel matrices into compact representations with statistical guarantees. However, existing analyses often fail to characterize the precise impact of sketching on uncertainty quantification and negative log marginal likelihood of the gaussian process regression. El Alaoui & Mahoney (2015) successfully applied the ridge leverage score based sampling technique to Nystrom approximation for kernel ridge regression. However it is non-trivial to extend their method to the case of GPR with provable guarantees , specifically for the predictive variance and negative log marginal likelihood approximation. Inspired by their method, we next describe the Nystrom Sketching for Kernel Approximation specifically for GPR and then describe our main contributions in this paper.

## 2 Nystrom Sketching for Kernel Approximation in GPR

To overcome the above computational challenges, we employ Nystrom sketching, which provides a low-rank approximation $\widehat{K}$ to $K$ while preserving its essential spectral structure.

**Nystrom approximation.** Let $J \subset \{1, \ldots, n\}$ be a set of $m \ll n$ sampled indices. Define

$$C = K_{:,J} \in \mathbb{R}^{n \times m}, \qquad W = K_{J,J} \in \mathbb{R}^{m \times m} \tag{5}$$

The classical Nystrom approximation is given by

$$\widehat{K} = C W^\dagger C^\top, \tag{6}$$

where $W^\dagger$ denotes the Moore–Penrose pseudoinverse of $W$. This approximation projects the kernel matrix onto the span of the sampled columns, yielding a rank-$m$ surrogate of $K$.

**Generalized sketching.** More generally, let $S \in \mathbb{R}^{n \times m}$ be a tall, skinny sketching matrix (e.g., sampling matrix, random projection, or structured transform). The Nystrom approximation associated with $S$ can be written as

$$\widehat{K}_S = K S (S^\top K S)^\dagger S^\top K, \tag{7}$$

which recovers equation 6 when $S$ corresponds to column sampling.

**Woodbury expansion for efficient solves.** In GPR, inference requires computing $(K + \sigma_\xi^2 I)^{-1}$, where $\sigma_\xi^2$ is the noise variance. Replacing $K$ with $\widehat{K}$ and applying the Woodbury identity 1 with $A = \sigma_\xi^2 I$, $U = C$, $M = W^{-1}$, $V = C^\top$, we obtain

$$(\widehat{K} + \sigma_\xi^2 I)^{-1} = \sigma_\xi^{-2} I - \sigma_\xi^{-4} C (W + \sigma_\xi^{-2} C^\top C)^{-1} C^\top \tag{8}$$

**Lemma 1** (Woodbury Nystrom Solve). *Let $\widehat{K} = C W^{-1} C^\top$ be the Nystrom approximation of $K$. Then for any $y \in \mathbb{R}^n$,*

$$(\widehat{K} + \sigma_\xi^2 I)^{-1} y = \sigma_\xi^{-2} y - \sigma_\xi^{-4} C (W + \sigma_\xi^{-2} C^\top C)^{-1} C^\top y \tag{9}$$

*Proof.* Apply the Woodbury matrix identity $(A + UMV)^{-1} = A^{-1} - A^{-1} U (M^{-1} + V A^{-1} U)^{-1} V A^{-1}$ with $A = \sigma_\xi^2 I$, $U = C$, $M = W^{-1}$, $V = C^\top$. □

This expression involves inverting only an $m \times m$ matrix, substantially reducing the computational burden.

**Predictive mean using Nystrom.** The GPR predictive mean for a test point $x_*$ with kernel vector $k_* \in \mathbb{R}^n$ is

$$\mu_* = k_*^\top (K + \sigma_\xi^2 I)^{-1} y \tag{10}$$

Using the Nystrom surrogate and equation 8, this becomes

$$\mu_* \approx \sigma_\xi^{-2} k_*^\top y - \sigma_\xi^{-4} k_*^\top C (W + \sigma_\xi^{-2} C^\top C)^{-1} C^\top y \tag{11}$$

Analogous derivations yield a similar reduction for the predictive variance.

**Computational complexity.** Constructing $C$ requires $\mathcal{O}(nm)$ kernel evaluations and storing it uses $\mathcal{O}(nm)$ memory. Forming and inverting $A = W + \sigma_\xi^{-2} C^\top C$ costs $\mathcal{O}(nm^2 + m^3)$, and each test prediction costs $\mathcal{O}(m^2)$. This is a dramatic improvement over the $\mathcal{O}(n^3)$ time and $\mathcal{O}(n^2)$ memory required for standard GPR.

**Our Contributions.** The main challenge in applying the sketched Nystrom method to GPR lies in designing an efficient sketching matrix and analyzing its impact on the predictive mean, predictive variance, and negative log-marginal likelihood (NLML). While sketching methods particularly those based on ridge leverage scores (RLS) have been well studied for Kernel Ridge Regression (KRR) El Alaoui & Mahoney (2015); Rasmussen & Williams (2005), their theoretical guarantees do not fully capture the unique properties of GPR. KRR analyses typically bound the statistical risk of the point predictor El Alaoui & Mahoney (2015), whereas GPR's strengths lie in uncertainty quantification via predictive variance and model selection through NLML Hensman et al. (2013). These quantities, fundamental to GPR, have no direct analogues in KRR; the GPR posterior variance is distinct from the variance of the KRR estimator, and NLML is critical for hyperparameter learning. Our work bridges this gap by extending guarantees for the predictive mean and, more importantly, providing the first explicit approximation bounds for the predictive variance and NLML under RLS sketching, establishing a complete theoretical foundation for scalable, high-fidelity GPR.

**Clarifying the scope of novelty.** While we adopt ridge leverage score (RLS) sampling as a kernel approximation primitive, our theoretical results do not follow from prior KRR analyses El Alaoui & Mahoney (2015). In contrast to KRR, GPR requires controlling predictive variance (inverse quadratic forms) and NLML (log-determinant terms), which are absent in prior work. Addressing these quantities necessitates new spectral perturbation arguments. Thus, our contribution provides the first theoretical guarantees for RLS-based sketching in full Bayesian GPR.

To summarize, the main contributions of this work are as follows:

1. We propose a kernel sketching framework based on ridge leverage scores for Gaussian Process Regression (GPR).

2. We provide, to the best of our knowledge, the first theoretical guarantees for ridge leverage score–based sketching specifically for the GPR problem. Specifically, we derive non-trivial bounds on the approximation error for the predictive mean, predictive variance, and negative log-likelihood.

3. We conduct extensive empirical evaluations across multiple real-world regression benchmarks, demonstrating the effectiveness of our method compared to standard baselines.

It is important to note that while our approach is comparable to some state-of-the-art methods in terms of runtime performance, we demonstrate that it achieves provable superior predictive quality and uncertainty calibration, thereby offering an accurate and efficient alternative to existing scalable GP techniques.

## 3 Related Work

Various methods have been applied to scale GPR for the big data regime; see Liu et al. (2020) and references therein. There are methods based on variational inference Hensman et al. (2013), and conjugate gradient–based iterative methods Artemev et al. (2021). However, handling GPR using sampling or sketching-based methods with theoretical guarantees is relatively less explored. Hayashi et al. (2020), using a novel graphon-based analysis, derive error bounds for Gaussian process subsampling via uniform random selection. However, their bounds decay slowly; for example, predictive error scales like $O(\log^{-1/4} s)$ with the number of subsamples $s$. Fiedler et al. (2021) provide practical bounds on GPR in general; however, these bounds are not directly comparable to ours.

*Nystrom approximation* Williams & Seeger (2001) reduces the computational complexity by projecting the full kernel matrix onto a subspace spanned by a set of inducing points. However, uniform or heuristic-based selection of these points often fails to capture critical data-dependent structure, especially in high-dimensional or non-uniform settings.

This has led to the adoption of more principled sampling techniques based on *ridge leverage scores* El Alaoui & Mahoney (2015), which offer spectral guarantees and have been successfully applied in kernel ridge regression Rudi & Rosasco (2015); Musco & Musco (2017) and randomized matrix approximation Drineas et al. (2012). Despite their theoretical appeal, ridge leverage based Nystrom approximations have not been

---

**Algorithm 1** $\sigma_\xi^2$-Ridge Leverage Score Sampling with Rescaling for Kernel Sketching

---

**Input:** Kernel matrix $K \in \mathbb{R}^{n \times n}$, noise parameter $\sigma_\xi^2 > 0$, sketch size $m \ll n$, Nystrom regularization parameter $\gamma > 0$
**Output:** Sketching matrix $S \in \mathbb{R}^{n \times m}$, sketched kernel $L_\gamma \in \mathbb{R}^{n \times n}$

1: Compute ridge leverage scores: $\ell_i^{(\sigma_\xi^2)} \leftarrow [K(K + \sigma_\xi^2 I)^{-1}]_{ii}$ for all $i \in \{1, \ldots, n\}$ ▷ See Algorithm 2 for fast computation
2: Normalize scores: $p_i \leftarrow \ell_i^{(\sigma_\xi^2)} / \sum_{j=1}^n \ell_j^{(\sigma_\xi^2)}$
3: Initialize $S \in \mathbb{R}^{n \times m}$ as a zero matrix

4: **for** $j = 1$ to $m$ **do**
5:     Sample index $i_j \sim \text{Categorical}(p_1, \ldots, p_n)$ ▷ Sample with replacement
6:     Set $S_{i_j,j} \leftarrow \frac{1}{\sqrt{m \cdot p_{i_j}}}$ ▷ Apply reweighting
7: **end for**

8: Compute sketched kernel: $L_\gamma \leftarrow KS(S^\top KS + \gamma I)^{-1} S^\top K$
9: **return** $S, L_\gamma$

---

widely explored in the context of Gaussian Process Regression particularly with respect to predictive quantities such as the posterior mean, variance, and marginal likelihood. Existing bounds are not directly applicable to GPR settings. In contrast, we leverage similar sampling strategies but develop new, explicit guarantees on these key predictive quantities, bridging this important gap.

Random Fourier Features (RFF) Rahimi & Recht (2007) approximate kernel functions using data-independent random feature maps, enabling scalable kernel learning without explicitly forming the kernel matrix. In contrast, Nystrom-based methods construct data-dependent low-rank approximations by selecting informative subsets of the data. While RFF can be effective when kernel evaluations are expensive, it typically requires storing dense $n \times D$ feature representations, often with large $D$ to achieve high accuracy. In comparison, Nystrom methods store $n \times m$ kernel factors with $m \ll n$, leading to a more compact representation. Moreover, since RFF does not adapt to the data distribution, it may require higher-dimensional features to accurately capture predictive uncertainty in Gaussian Process settings. In contrast, ridge leverage score–based Nystrom sketching prioritizes statistically informative points, resulting in improved approximation of predictive variance and marginal likelihood, particularly in settings where memory efficiency and uncertainty calibration are critical.

## 4 Algorithms

In this section, we outline the algorithms used to sketch the kernel matrix. While similar techniques have been studied in kernel ridge regression El Alaoui & Mahoney (2015), their application to Gaussian Process Regression (GPR) with explicit theoretical guarantees has not been previously established.

We use a generalized notion of leverage scores specifically designed for the ridge regression setting, referred to as the $\sigma_\xi^2$-*ridge leverage scores*.

**Definition 1.** *Given $\sigma_\xi^2 > 0$, the $\sigma_\xi^2$-ridge leverage scores corresponding to a kernel matrix $K$ and regularization/noise parameter $\sigma_\xi^2$ are defined as*

$$\forall i \in \{1, \ldots, n\}, \quad l_i(\sigma_\xi^2) = \sum_{j=1}^n \frac{\sigma_j}{\sigma_j + \sigma_\xi^2} U_{ij}^2$$

Here, $l_i(\sigma_\xi^2)$ represents the $i^{\text{th}}$ diagonal entry of the matrix product $K(K + \sigma_\xi^2 I)^{-1}$, where $\sigma_j$ denotes the $j^{\text{th}}$ eigenvalue of the kernel matrix $K$, and $U$ is the orthonormal matrix of eigenvectors from its eigendecomposition. The set $(l_i(\sigma_\xi^2))_{1 \leq i \leq n}$ serves a similar role to classical leverage scores in statistics, as they help identify

---

**Algorithm 2** Approximate $\sigma_\xi^2$-Ridge Leverage Score Computation via Nystrom Sketching El Alaoui & Mahoney (2015)

---

    **Input:** Data points $\{x_1, \ldots, x_n\}$, kernel function $k(\cdot, \cdot)$, sampling distribution $\{p_i\}_{i=1}^n$, sketch size $m$, regularization parameter $\sigma_\xi^2 > 0$

    **Output:** Approximate ridge leverage scores $\{\tilde{\ell}_i\}_{i=1}^n$

1: Sample indices $i_1, \ldots, i_m \sim \text{Categorical}(p_1, \ldots, p_n)$ with replacement
2: Form matrix $C \in \mathbb{R}^{n \times m}$ such that $C_{j,\ell} = k(x_j, x_{i_\ell})$
3: Form $W \in \mathbb{R}^{m \times m}$ with $W_{\ell,p} = k(x_{i_\ell}, x_{i_p})$
4: Compute $B \in \mathbb{R}^{n \times m}$ such that $BB^\top = CW^\dagger C^\top$                ▷ Can use Cholesky or QR on $W$
5: Compute matrix $M = (B^\top B + \sigma_\xi^2 I_m)^{-1}$
6: **for** $i = 1$ to $n$ **do**
7:     Set $\tilde{\ell}_i \leftarrow B_i^\top M B_i$                                   ▷ $B_i$ is the $i$-th row of $B$
8: **end for**
9: **return** $\{\tilde{\ell}_i\}_{i=1}^n$

---

influential data points that significantly impact the model output. In traditional settings, these scores are often derived from the row norms of the left singular vectors in the matrix $U$.

The effective dimension, denoted by $d_{\text{eff}}(\sigma_\xi^2)$, is defined as

$$d_{\text{eff}}(\sigma_\xi^2) = \text{Tr}\left(K(K + \sigma_\xi^2 I)^{-1}\right)$$

where $K$ is the kernel matrix and $\sigma_\xi^2 > 0$ is the regularization/noise parameter.

To efficiently approximate the $\sigma_\xi^2$-ridge leverage scores without computing the full eigendecomposition of the kernel matrix, we adopt an approximation strategy inspired by Algorithm 2 El Alaoui & Mahoney (2015).

The approximation algorithm 2 accepts as input a sampling distribution over the data points, which we set to a simple yet effective diagonal proxy where each point is sampled with probability proportional to the diagonal entry of the kernel matrix, i.e., $p_i = \frac{K_{ii}}{\text{Tr}(K)}$. This choice is motivated by the fact that the diagonal of $K$ captures the self-similarity of each point and offers a computationally efficient surrogate for ridge leverage scores. The algorithm then selects a subset of size $m$, computes the corresponding kernel submatrices $C$ and $W$, and uses a Nyström-style factorization to produce an approximate low-rank embedding $B$. The resulting approximate leverage scores are given by the quadratic form $\tilde{\ell}_i = B_i^\top (B^\top B + \sigma_\xi^2 I)^{-1} B_i$ for each point $i$. This method has runtime $\mathcal{O}(nm^2)$ and storage complexity $\mathcal{O}(nm)$, and provides provably accurate approximations to the true ridge leverage scores with high probability, while being scalable to large datasets where exact score computation is infeasible.

## 5 Theoretical Guarantees

In this section, we present our main theoretical results for the predictive mean (Theorem 2), variance (Theorem 3), and negative log-likelihood (Theorem 4) under ridge leverage score–based sketching. We begin by outlining the setup, notation, and assumptions common to all three theorems. For better readability, we have deferred detailed proofs to the appendix.

**Setup:** Let $\mathcal{D} = \{(x_i, y_i)\}_{i=1}^n$ be the dataset, where $x_i \in \mathbb{R}^d$ are the input points and $y_i \in \mathbb{R}$ are the corresponding outputs. Let $k(x, x')$ be the kernel function used in the Gaussian process regression, and let $K$ be the $n \times n$ kernel matrix such that $K_{ij} = k(x_i, x_j)$. Let the eigenvalue decomposition of the kernel matrix be $K = U\Sigma U^T$.

Let $S \in \mathbb{R}^{n \times m}$ be a sketching matrix (obtained in Algorithm 1) so that $S_{ij} = \sqrt{\frac{1}{mp_i}}$ if $i = i_j$ else 0, where $m \ll n$ is obtained by probability distribution $(p_i)_{1 \leq i \leq n}$ such that $\forall i \in \{1, \cdots, n\}$, $\quad p_i \geq \beta \cdot l_i(\sigma_\xi^2)/\sum_{i=1}^n l_i(\sigma_\xi^2)$ for some $\beta \in (0, 1]$. We define the sketch of a kernel matrix $L_\gamma = KS(S^\top KS + \gamma I)^{-1}S^\top K$ as the submatrix of $K$ where $\gamma > 0$.

Moreover, let

$$D = \Phi - \Phi^{1/2}U^\top SS^\top U\Phi^{1/2}$$

with $\Phi = \Sigma(\Sigma + \gamma I)^{-1}$.

From, El Alaoui & Mahoney (2015) we have that, as long as the sketching matrix $S$ satisfies $\lambda_{max}(D) \le t$ for $t \in (0,1)$ and $\lambda_{max}$ denoting the maximum eigenvalue. Under this condition, they show that the approximation error between the original kernel matrix $K$ and its sketched version $L_\gamma$ is bounded as

$$0 \preceq K - L_\gamma \preceq \left(\frac{\gamma}{1-t}\right)I$$

To ensure this bound holds, we follow the **sketch size guarantee** provided in Theorem 2 of Appendix B in El Alaoui & Mahoney (2015), which characterizes the **required number of samples** $m$ based on the ridge leverage score distribution is,

$$m \ge 8\left(\frac{d_{\text{eff}}}{\beta} + \frac{1}{6}\right)\log\left(\frac{n}{\delta}\right)$$

For the mean and variance inference on test data, we express the test kernel vector as $k_* = U\alpha$, where $\alpha \in \mathbb{R}^n$ with $\alpha_i = u_i^\top k_*$, and $U = [u_1, \dots, u_n]$ is the eigenvector matrix of $K$. This is a standard change-of-basis in the eigenspace of $K$ and does not introduce any additional modeling assumptions.

**Interpretation of $\alpha$ and $\beta$.** The coefficients $\alpha$ quantify the alignment of the test point with the spectral components of the kernel matrix, and determine how the eigenvalue spectrum influences the approximation error.

The parameter $\beta \in (0,1]$ controls the quality of the sampling distribution relative to the ridge leverage scores through the condition $p_i \ge \beta\,\ell_i/\sum_j \ell_j$. Larger $\beta$ corresponds to more accurate leverage-based sampling (e.g., exact leverage scores when $\beta = 1$), while smaller $\beta$ allows approximate scores and leads to a larger required sketch size $m = \tilde{O}(d_{\text{eff}}/\beta)$.

Our analysis conditions on the observed dataset and quantifies uncertainty solely with respect to the randomness of the sketching procedure. This follows the standard paradigm in randomized numerical linear algebra Mahoney (2011); Pilanci & Wainwright (2015); Musco & Musco (2017), where sketching is viewed as a computational approximation mechanism rather than a statistical estimator of the data-generating process. From a practical perspective, this form of guarantee is particularly relevant: given a fixed dataset, our results ensure that the sketched GPR solution closely approximates the exact solution with high probability over the sketch. This interpretation aligns with prior work in kernel sketching and randomized matrix methods.

## 5.1 Predictive Mean Estimation using Sketching

**Theorem 2** (Predictive Mean Approximation under $\sigma_\xi^2-$ Ridge Leverage Score Sketching). *For the notations and assumptions defined in our* **Setup** *let*

$$\mu(x^*) = k_*^T(K + \sigma_\xi^2 I)^{-1}y$$

*be the predictive mean of Gaussian Process Regression at a new point $x^*$ for the full kernel matrix. Here $k_* = [k(x_1,x^*), k(x_2,x^*), \dots, k(x_n,x^*)]^T$ and $\sigma_\xi^2$ is the noise. For the sketch of a kernel matrix $L_\gamma$, the predictive mean is,*

$$\mu_S(x^*) = k_*^T(L_\gamma + \sigma_\xi^2 I)^{-1}y$$

*For the $L_\gamma$ obtained using Algorithm 1 we have*

$$|\mu(x^*) - \mu_S(x^*)| \le \left(\frac{\gamma}{1-t}\right)\sqrt{\sum_{i=1}^n \frac{\alpha_i^2}{(\Sigma_{i,i} + \sigma_\xi^2)^2}} \cdot \|y\|_2 \cdot \lambda_{max}(\Delta_D)$$

*where,* $\Delta_D = \left( \Sigma \left[ I - \frac{\gamma}{1-t}(\Sigma + \gamma I)^{-1} \right] + \sigma_\xi^2 I \right)^{-1}$, *hold with probability at least* $1 - \delta$, *if the sketch size* $m$ *is set so that*

$$m \geq 8 \left( \frac{d_{eff}}{\beta} + \frac{1}{6} \right) \log \left( \frac{n}{\delta} \right)$$

**Remark (Interpretation of $\alpha$ in the Predictive Mean and Variance Bounds).** In the predictive mean approximation bound, the term involving $\alpha_i^2$ arises from expressing the test-to-train kernel vector $k_* \in \mathbb{R}^n$ in the eigenbasis of the kernel matrix $K = U\Sigma U^\top$, such that $k_* = U\alpha$ with $\alpha = U^\top k_*$. The coefficients $\alpha_i$ quantify the alignment of the test point $x_*$ with the spectral components of the training kernel and thus determine how the eigenvalue spectrum of $K$ influences the approximation error.

**Discussion.** The bound depends on the energy of the test kernel vector in the eigenbasis of $K$, captured by $\sum_i \alpha_i^2/(\Sigma_{ii} + \sigma_\xi^2)^2$. This term directly links the approximation quality to both the spectral decay of the kernel and the geometric relation between the test and training points. When the kernel spectrum decays rapidly such as for smooth kernels like RBF or high-order Matern the contributions from low-eigenvalue directions are strongly attenuated, leading to tighter bounds. Similarly, a larger noise variance $\sigma_\xi^2$ regularizes the influence of small eigenvalues, further stabilizing the approximation. Hence, the derived error bounds characterize how spectral compressibility of the kernel governs the fidelity of the sketched approximation without imposing additional assumptions on the distribution of the test inputs. An analogous interpretation applies to the predictive variance bound (Theorem 3), where the vector $\alpha$ again captures the projection of the test point onto the eigenspace of the kernel matrix.

We now specialize our predictive mean approximation to a standard nonparametric setting with polynomial spectral decay and a source condition. These assumptions capture the smoothness of the underlying function and the effective complexity of the kernel through its eigenvalue decay.

Under this regime, our bounds translate into explicit learning rates that describe how the predictive mean error scales with sample size and noise. In particular, ridge leverage score–based Nystrom sketching achieves near-optimal rates (up to logarithmic factors) while using only a sublinear number of inducing points governed by the effective dimension.

**Corollary 1** (Predictive Mean Approximation under Polynomial Spectrum and Source Condition). *Let* $K = U\Sigma U^\top$ *with eigenvalues satisfying polynomial decay*

$$\sigma_i \asymp i^{-\rho}, \qquad \rho > 1,$$

*and assume the test kernel vector satisfies the source condition of order* $r > 1/2$:

$$k_* = U\alpha, \qquad \alpha_i^2 \leq C\,\sigma_i^{1+2r}, \quad i = 1, \ldots, n.$$

*Assume the Nystrom sketch satisfies the ridge leverage condition with*

$$m \gtrsim d_{\text{eff}}(\sigma_\xi^2) \log(n/\delta), \qquad d_{\text{eff}}(\lambda) = \text{Tr}\big(K(K + \lambda I)^{-1}\big),$$

*and that* $\gamma \asymp \sigma_\xi^2$.

*Let* $\sigma_\xi^2 \asymp n^{-\beta}$ *for* $\beta > 0$. *Then with probability at least* $1 - \delta$,

$$|\mu(x_*) - \mu_S(x_*)| = O\Big( \sigma_y\, n^{-\beta\left(r - \frac{1}{2} - \frac{1}{2\rho}\right)} \Big), \qquad m = O\Big( n^{\beta/\rho} \log n \Big),$$

*where* $\sigma_y^2 = \mathbb{E}[\|y\|_2^2]$ *is the expected squared norm of the observation vector.*

*In particular, for the standard scaling* $\sigma_\xi^2 \asymp n^{-1}$ *(i.e.,* $\beta = 1$*), this yields*

$$|\mu(x_*) - \mu_S(x_*)| = O\Big( \sigma_y\, n^{-\left(r - \frac{1}{2} - \frac{1}{2\rho}\right)} \Big), \qquad m = O\Big( n^{1/\rho} \log n \Big)$$

We now strengthen our predictive mean approximation result by establishing a uniform guarantee over the entire input domain. While previous bounds control the error at a fixed test point, uniform bounds are essential for ensuring that the approximation remains stable across all possible test inputs.

Under mild assumptions on kernel boundedness and sketching parameters, we obtain a high-probability bound on the worst-case predictive mean error over the domain. This result shows that the Nystrom approximation preserves the predictive behavior of the full Gaussian process uniformly, providing stronger guarantees for downstream tasks that require consistent performance across the input space.

**Corollary 2** (Uniform Predictive Mean Approximation). *Assume that the kernel is uniformly bounded on a compact domain $\mathcal{X}$, i.e.,*

$$\sup_{x \in \mathcal{X}} k(x, x) \leq \kappa^2$$

*and that the sketching parameters satisfy*

$$\sigma_\xi^2 > \frac{t\gamma}{1 - t + t}$$

*Then, with probability at least $1 - \delta$ over the sketching randomness, the predictive mean error satisfies*

$$\sup_{x^* \in \mathcal{X}} |\mu(x^*) - \mu_S(x^*)| \leq \frac{\gamma \kappa}{(1 - t)\sqrt{\sigma_\xi^2}} \cdot \frac{\|y\|_2}{\sigma_\xi^2 - \frac{t\gamma}{1 - t + t}}$$

### 5.2 Predictive Variance Estimation using Sketching

**Theorem 3** (Predictive Variance Approximation under $\sigma_\xi^2-$ Ridge Leverage Score Sketching). *For the notations and assumptions defined in our **Setup** let*

$$Var(x^*) = k(x^*, x^*) - k_*^\top (K + \sigma_\xi^2 I)^{-1} k_*$$

*be the predictive variance of Gaussian Process Regression at a new point $x^*$. Here, $k_* = [k(x_1, x^*), k(x_2, x^*), \ldots, k(x_n, x^*)]^T$ and $\sigma_\xi^2$ is the noise variance. For the sketch of a kernel matrix $L_\gamma$, the predictive variance is,*

$$Var_S(x^*) = k(x^*, x^*) - k_*^\top (L_\gamma + \sigma_\xi^2 I)^{-1} k_*$$

*For the $L_\gamma$ obtained using Algorithm 1 we have*

$$|Var(x^*) - Var_S(x^*)| \leq \left(\frac{\gamma}{1 - t}\right) \|\alpha^\top \Delta_D\|_2 \cdot \sqrt{\sum_{i=1}^n \alpha_i^2 \left(\frac{1}{(\Sigma_{i,i} + \sigma_\xi^2)}\right)^2}$$

*where, $\Delta_D = \left(\Sigma \left[I - \frac{\gamma}{1-t}(\Sigma + \gamma I)^{-1}\right] + \sigma_\xi^2 I\right)^{-1}$, hold with probability at least $1 - \delta$ if the sketch size $m$ is set so that*

$$m \geq 8 \left(\frac{d_{eff}}{\beta} + \frac{1}{6}\right) \log\left(\frac{n}{\delta}\right)$$

We now extend the uniform analysis to predictive variance, which plays a central role in uncertainty quantification in Gaussian process regression. While pointwise guarantees control the variance approximation at a fixed test input, a uniform bound ensures that uncertainty estimates remain reliable across the entire input domain.

Under the same boundedness and sketching assumptions, we obtain a high-probability bound on the worst-case predictive variance error. This result shows that Nystrom sketching preserves not only the predictive mean but also the uncertainty estimates uniformly, providing a principled guarantee for downstream tasks that rely on well-calibrated uncertainty.

**Corollary 3** (Uniform Predictive Variance Approximation). *Assume that the kernel is uniformly bounded on a compact domain $\mathcal{X}$, i.e.,*

$$\sup_{x \in \mathcal{X}} k(x,x) \leq \kappa^2$$

*and that the sketching parameters satisfy*

$$\sigma_\xi^2 > \frac{t\gamma}{1-t+t}$$

*Then, with probability at least $1-\delta$ over the sketching randomness, the predictive variance error satisfies*

$$\sup_{x^* \in \mathcal{X}} \left| \mathrm{Var}(x^*) - \mathrm{Var}_S(x^*) \right| \leq \frac{\gamma \kappa^3 \sqrt{n}}{(1-t)\sqrt{\sigma_\xi^2}} \cdot \frac{1}{\sigma_\xi^2 - \frac{t\gamma}{1-t+t}}$$

### 5.3 Negative Log Marginal Likelihood Approximation

**Theorem 4** (Negative Log Marginal Likelihood Approximation under $\sigma_\xi^2-$ Ridge Leverage Score Sketching). *Let $K \in \mathbb{R}^{n \times n}$ be a symmetric positive semi-definite kernel matrix and $y \in \mathbb{R}^n$ the response vector. For $\sigma_\xi^2 > 0$, the negative log marginal likelihood (NLML) be,*

$$\mathcal{L}(K) = \frac{1}{2} y^\top (K + \sigma_\xi^2 I)^{-1} y + \frac{1}{2} \log \det(K + \sigma_\xi^2 I) + \frac{n}{2} \log(2\pi)$$

*The corresponding approximate NLML for the sketch of the kernel matrix $L_\gamma$ obtained using Algorithm 1 is given as,*

$$\mathcal{L}(L_\gamma) = \frac{1}{2} y^\top (L_\gamma + \sigma_\xi^2 I)^{-1} y + \frac{1}{2} \log \det(L_\gamma + \sigma_\xi^2 I) + \frac{n}{2} \log(2\pi)$$

*Then, for any $0 \leq \delta \leq 1$, if*

$$m \geq 8 \left( \frac{d_{eff}}{\beta} + \frac{1}{6} \right) \log \left( \frac{n}{\delta} \right)$$

*then, with probability at least $1-\delta$ following inequality holds,*

$$|\mathcal{L}(K) - \mathcal{L}(L_\gamma)| \leq \frac{\gamma}{2(1-t)} \left( \frac{1}{\lambda_{min}(K + \sigma_\xi^2 I)} \right) \cdot \|y\|_2^2 \cdot \lambda_{max}(\Delta_D) + \frac{\gamma}{2(1-t)} \mathrm{Tr}(\Delta_D)$$

*where, $\Delta_D = \left( \Sigma \left[ I - \frac{\gamma}{1-t}(\Sigma + \gamma I)^{-1} \right] + \sigma_\xi^2 I \right)^{-1}$.*

We now turn to the approximation of the negative log marginal likelihood (NLML), which governs hyperparameter learning and model selection in Gaussian process regression. Unlike predictive quantities, NLML is a global scalar functional of the kernel matrix and does not depend on a test input.

Under the sketching assumptions, we obtain a high-probability bound on the NLML approximation error, showing that Nystrom sketching preserves the training objective up to a controlled perturbation. This provides a principled guarantee that sketching not only maintains predictive accuracy but also faithfully approximates the objective used for learning kernel hyperparameters.

**Corollary 4** (Uniform NLML Approximation). *Assume that the sketching parameters satisfy*

$$\sigma_\xi^2 > \frac{t\gamma}{1-t+t}$$

*Then, with probability at least $1-\delta$ over the sketching randomness, for all $y \in \mathbb{R}^n$,*

$$\left| \mathcal{L}(K) - \mathcal{L}(L_\gamma) \right| \leq \frac{\gamma}{2(1-t)} \left[ \frac{\|y\|_2^2}{\sigma_\xi^2} \cdot \frac{1}{\sigma_\xi^2 - \frac{t\gamma}{1-t+t}} + \sum_{i=1}^{n} \frac{1}{\sigma_i - \frac{t\gamma}{1-t+t} + \sigma_\xi^2} \right]$$

### 5.4 Interpretation of $\Delta_D$

All bounds in our analysis share a central spectral term involving the matrix

$$\Delta_D = \left( \Sigma \left[ I - \frac{\gamma}{1-t}(\Sigma + \gamma I)^{-1} \right] + \sigma_\xi^2 I \right)^{-1}$$

where $\Sigma$ denotes the diagonal matrix of kernel eigenvalues, $\gamma > 0$ is the regularization parameter, $t \in (0,1)$, and $\sigma_\xi^2$ is the noise.

**Spectral Dependence of the Bounds.** The tightness of the predictive mean, variance, and NLML bounds is depends on the terms $\lambda_{\max}(\Delta_D)$ and $\text{Tr}(\Delta_D)$, both of which are minimized when the spectrum of $\Sigma$ exhibits fast decay. In such cases, $\Delta_D$ becomes better conditioned, as low-eigenvalue directions are strongly regularized or suppressed by the additive noise. This yields tighter theoretical guarantees for the sketched approximation. Smooth kernels such as the RBF and high-$\nu$ Matérn families naturally induce this spectral decay, particularly when applied to well-distributed, low-dimensional inputs typical of geostatistical data or physical simulations.

## 6 Experiments

All experiments were conducted on a machine equipped with an NVIDIA A100 PCIe GPU with 32 GB of memory. Our implementation is written in Python and leverages PyTorch and GPyTorch Gardner et al. (2018) for efficient GPU-accelerated Gaussian Process modeling.

### 6.1 Datasets

We evaluate our methods on four real-world regression datasets: `California Housing` Pace & Barry (1997), `Elevators` Team (1996), `Airfoil Self-Noise` H. et al. (1999), and `Protein` Cai et al. (2003). All datasets are standardized using z-score normalization for both inputs and targets. For `California Housing` and `Protein`, we use a 70%/30% train/test split; for the others, we follow an 80%/20% split.

`California Housing` contains 20,640 samples with 8 real-valued features describing demographic and geographic attributes from the 1990 U.S. Census. The target variable is the median house value in each district, making it a widely used benchmark for medium-scale regression tasks with heterogeneous feature distributions.

`Elevators` is a large-scale regression benchmark from the DELVE framework, hosted on the UCI repository. It consists of 16,599 samples with 18 continuous features capturing the dynamics of a control system, and a real-valued target representing elevator response time. The dataset exhibits moderately complex and nonlinear patterns, making it well-suited for testing scalable GP models.

`Airfoil Self-Noise` comprises 1,503 samples with 5 continuous features representing physical properties and operating conditions of airfoils in a wind tunnel. The target is the scaled sound pressure level. Due to its small size and nonlinear behavior, it serves as a testbed for evaluating predictive uncertainty in low-data regimes.

`Protein` (also known as `Protein Structure`) is a large-scale regression dataset from the UCI repository with 45,730 samples and 9 physicochemical features describing the secondary structure of proteins. The target variable is the root mean square deviation (RMSD) of atomic positions, which measures structural variability. This dataset is widely used to benchmark scalable kernel methods due to its size, moderate dimensionality, and nonlinear structure.

### 6.2 Experimental Setup

We use the Radial Basis Function (RBF) kernel and Matern kernel for all three datasets. The kernel hyperparameters, including the lengthscales and variance, are initialized to 1.0 in case of RBF kernel and in

Matern kernel $\nu = 1.5$ is initialized. The prior mean function is initialized as a constant set to 0 and is treated as a learnable hyperparameter during training. Based on preliminary experiments, we fix the learning rate to 0.01 and train for 300 iterations across all methods. In contrast, SVGP was trained for 1000 iterations. This is because SVGP, as a variational method, optimizes an objective (the ELBO) that iteratively approximates the true posterior, a process that generally requires more iterations to stabilize than the methods that optimize the exact marginal log-likelihood. We trained SVGP using Adam (lr=0.01) and mini-batch size 1024.

## 6.3 Baselines

We compare our Nystrom Ridge Leverage method against a comprehensive set of baselines spanning both the coreset selection and scalable GPR literature which are described below. All methods are evaluated over progressively increasing subset sizes, covering approximately 2% to 12% of the full training set. To ensure statistical robustness and account for variability in subset construction, each experiment is repeated across 5 independent random seeds, where each seed corresponds to a different data split and independently selected subset. We report the mean metric values across these trials, along with the corresponding standard deviations to reflect variability and robustness.

**Uniform Subsampling.** Uniform subsampling Hayashi et al. (2020); Malaviya et al. (2024) selects training points uniformly at random, independent of the data distribution or kernel structure. Although simple and computationally efficient, it often fails to capture important geometric or uncertainty-related aspects of the data.

**Leverage Score Sampling.** Leverage score sampling prioritizes points with higher statistical influence, emphasizing those contributing most to the low-rank structure of the kernel matrix Drineas et al. (2012); Zheng & Phillips (2017); Chhaya et al. (2020). This data-aware selection improves representativeness over uniform sampling and provides a foundation for more advanced sketching-based approaches.

**$k$-Means Coreset.** We include a $k$-means-based coreset baseline using the Lightweight Coreset method Bachem et al. (2018); Shit et al. (2022), which combines uniform and sensitivity-based sampling to select representative points with replacement. This efficiently approximates the data's clustering structure and scales better than exact $k$-means on large datasets.

**Stochastic Variational Gaussian Processes (SVGP).** SVGP Hensman et al. (2013) is a variational inference framework for scalable GPR that optimizes an evidence lower bound (ELBO) via stochastic gradients. It supports mini-batch training and inducing point learning, and is widely regarded as a state-of-the-art method for large-scale Gaussian Processes.

**IterGP.** IterGP (Wenger et al., 2022) introduces a computation-aware framework for Gaussian Process inference that explicitly models both *mathematical uncertainty* (due to finite data) and *computational uncertainty* (due to approximate inference). Unlike standard approximations such as SVGP or CG-based solvers that ignore the uncertainty introduced by limited compute, IterGP provides a combined posterior whose covariance decomposes into mathematical and computational components. This guarantees convergence of the posterior mean in RKHS norm and offers a worst-case bound on the approximation error.

**Random Fourier Features (RFF).** Random Fourier Features (RFF) approximate shift-invariant kernels using randomized feature maps, enabling scalable learning without explicitly forming the kernel matrix. The kernel is approximated via a finite-dimensional feature embedding of size $D$, followed by linear inference. In our experiments, we use $D = 512$ features. However, being data-independent, RFF typically requires large $D$ to achieve high accuracy in Gaussian Process settings Rahimi & Recht (2007).

**Nystrom Approximations.** We compare three Nystrom based kernel approximation methods that differ in how the inducing points (columns of the kernel matrix) are selected. The uniform variant samples columns uniformly at random with replacement Williams & Seeger (2001), while the leverage score variant uses sampling probabilities proportional to the standard leverage scores Gittens & Mahoney (2016). Our method

employs ridge leverage score sampling, which incorporates the regularization parameter and provides a data-aware, theoretically grounded alternative for constructing Nystrom approximations in GPR.

## 6.4 Evaluation Metrics

Model performance is evaluated using the following metrics: (i) predictive mean error, (ii) predictive variance error, (iii) root mean squared error (RMSE), (iv) Negative Log Predictive Density (NLPD), (v) Mean Standardized Log Loss (MSLL), and (vi) Negative log likelihood (NLL). Predictive mean and variance errors are computed as the relative $\ell_2$ norm difference with respect to the full Gaussian Process model, defined as $\|\mu_{\text{full}} - \mu_{\text{sketch}}\|/\|\mu_{\text{full}}\|$ and $\|\sigma^2_{\text{full}} - \sigma^2_{\text{sketch}}\|/\|\sigma^2_{\text{full}}\|$, respectively.

**Negative Log Predictive Density (NLPD)**   To evaluate the quality of uncertainty estimates in Gaussian Process Regression (GPR), we report the Negative Log Predictive Density (NLPD). NLPD measures how well the predicted Gaussian distribution aligns with the true targets, penalizing both misestimated means and variances. Formally, for test data $\{(x_i, y_i)\}_{i=1}^n$ with predictive mean $\mu_i$ and variance $\sigma_i^2$, NLPD is computed as:

$$\text{NLPD} = \frac{1}{n} \sum_{i=1}^n \left( \frac{(y_i - \mu_i)^2}{2\sigma_i^2} + \frac{1}{2} \log(2\pi\sigma_i^2) \right) \tag{12}$$

Lower values indicate better predictive performance and better-calibrated uncertainty. NLPD is a proper scoring rule and is widely used in evaluating probabilistic regression models Artemev et al. (2021); Rasmussen & Williams (2005).

**Mean Standardized Log Loss (MSLL).**   Unlike standard error metrics such as RMSE, MSLL evaluates how well the predictive distribution improves over a simple baseline model (typically the empirical mean and variance of the training targets). Formally, MSLL is defined as

$$\text{MSLL} = \frac{1}{n} \sum_{i=1}^n \left[ \log p(y_i \mid x_i, \mathcal{D}_{\text{train}}) - \log p_{\text{baseline}}(y_i \mid x_i) \right], \tag{13}$$

where $p(y_i \mid x_i, \mathcal{D}_{\text{train}})$ denotes the model's predictive density and $p_{\text{baseline}}(y_i \mid x_i)$ corresponds to the baseline predictive distribution. A negative MSLL indicates that the model outperforms the baseline in terms of log predictive density. MSLL is particularly useful because it standardizes performance across datasets with different output scales and provides a more interpretable measure of probabilistic performance than raw log likelihood (Rasmussen & Williams, 2005).

## 6.5 Results and Analysis

Our main results, presented in the figures below and detailed in the tables in the appendices, demonstrate that the proposed Nystrom ridge leverage sketching method consistently outperforms all considered baselines across datasets, most notably on strong metrics such as NLPD and MSLL. Despite SVGP and IterGp being a state-of-the-art approach for scalable GPs, particularly in probabilistic modeling, our method achieves superior performance, especially in terms of Negative Log Predictive Density (NLPD), which is a proper scoring rule sensitive to both prediction accuracy and uncertainty calibration. Our method also achieves consistently lower predictive mean and variance errors, indicating that it more accurately approximates the true posterior distribution of the Gaussian Process Regression. Notably, the NLPD gains are achieved using an efficient, approximate version of ridge leverage score computation Algorithm 2, showcasing the scalability and effectiveness of our approach. The best-performing results are shown in bold. We focus on these representative baselines to cover the most widely used paradigms for scalable GPR subset selection, Nystrom approximation, iterative approximation, and variational inference.

While several variational approaches to scalable Gaussian Process inference exist, we include SVGP as it remains the most widely adopted and well-established representative of this class. Additionally, we compare against IterGP, a recent state-of-the-art scalable GPR method, to benchmark our approach against the strongest contemporary baselines.

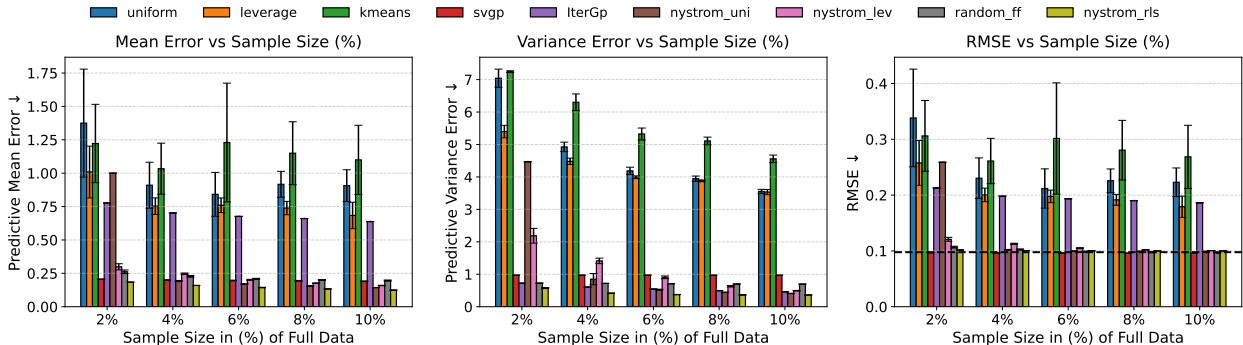

Figure 1: **Results on UCI Elevators Dataset.** Evaluation of Gaussian Process Regression methods on the UCI Elevators dataset using the **RBF kernel**. Predictive mean error, predictive variance error, and RMSE are plotted versus subset size. Ridge Leverage based GPR yields the best tradeoff across metrics. All results are averaged over 5 random trials with standard deviation shown as error bars. The dashed horizontal line indicates the performance of the full-dataset (exact GP) model.

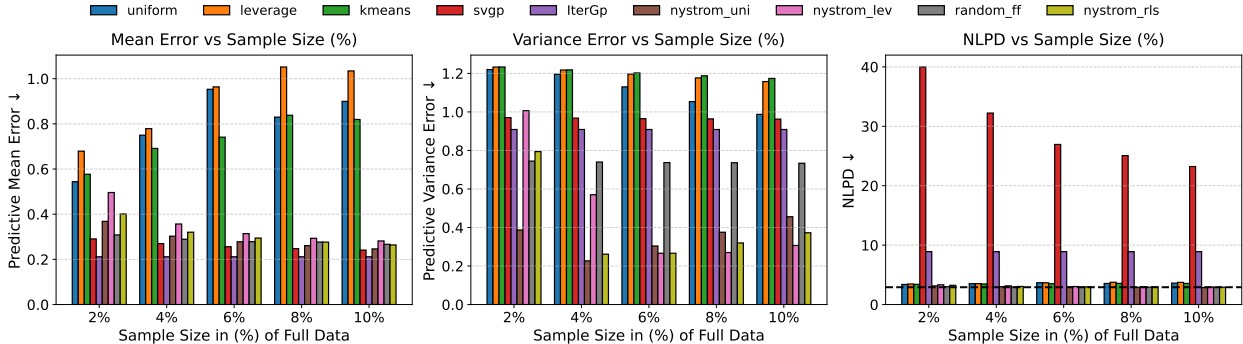

Figure 2: **Results on Protein Dataset.** Performance comparison of various Gaussian Process Regression (GPR) methods on the `Protein` dataset using the **RBF kernel**. The ridge leverage−based sketching method achieves superior predictive variance and NLPD compared to uniform, IterGp, SVGP, RFF and other baselines, demonstrating its robustness on this high-dimensional, large-scale regression task. The reported NLPD and uncertainty metrics are averaged over 5 random trials, with error bars representing standard deviations across runs. The dashed horizontal line indicates the performance of the full-dataset (exact GP) model.

For the California Housing dataset, we were unable to include the Nystrom (leverage) baseline, as the resulting kernel matrix approximation was not positive semi-definite, which caused instability during model training. More results are included in the appendix.

**Training Time and Dataset Scale Justification.** In our experiments, the SVGP baseline often required *longer training time* than exact GPR despite its theoretical scalability. This effect is prominent for moderate-scale datasets ($n \approx 15K$ to $30K$), as GPyTorch's exact GPR leverages efficient conjugate gradient routines and optimizes only a few kernel hyperparameters, whereas SVGP jointly learns kernel and variational parameters through stochastic updates. Moreover, mini-batching and stochastic optimization introduce additional overhead at this scale, making SVGP slower in wall-clock time compared to the exact solver for moderate $n$, a behavior also observed in prior work (Wilson & Nickisch, 2015; Gardner et al., 2018; Pleiss et al., 2018). We therefore restrict our benchmark to the Protein dataset ($n = 45,730$) the largest size for which full GPR remains tractable on a 32 GB GPU. Beyond this scale, storing and inverting the full kernel ($O(n^3)$ time, $O(n^2)$ memory) becomes infeasible, preventing computation of reference quantities such as predictive mean or variance errors. This regime allows meaningful and fair comparison against scalable methods while maintaining exact GPR as a ground-truth reference (Rasmussen & Williams, 2005; Gardner et al., 2018; Wang et al., 2019).

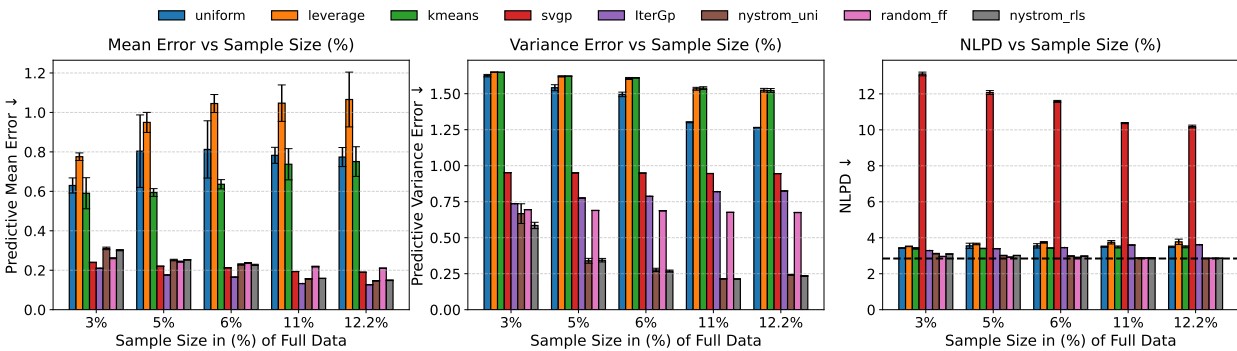

Figure 3: **Results on Protein Dataset.** Performance comparison of various Gaussian Process Regression (GPR) methods on the `Protein` dataset using the **Matern kernel**. The ridge leverage−based sketching method achieves superior predictive variance and NLPD compared to uniform, IterGp, SVGP, RFF and other baselines, demonstrating its robustness on this high-dimensional, large-scale regression task. The reported NLPD and uncertainty metrics are averaged over 5 random trials, with error bars representing standard deviations across runs. The dashed horizontal line indicates the performance of the full-dataset (exact GP) model.

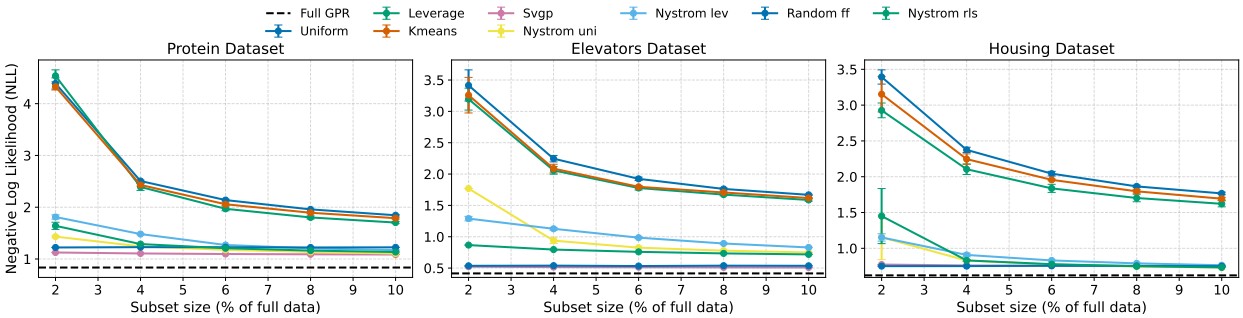

Figure 4: Comparison of Negative Log Likelihood (NLL) across different subset sizes and Gaussian Process approximation methods on the Protein, Elevators, and Housing datasets using the RBF kernel. Each subplot reports the mean and standard deviation over five random trials. The dashed horizontal line denotes the performance of the full-data (Exact GP) model.

# 7 Conclusion

We proposed a scalable Gaussian Process Regression method that combines Nystrom approximation with ridge leverage score sampling. While ridge leverage scores have been used in kernel ridge regression and matrix approximation, our work is the first to apply them in the Gaussian Process setting with theoretical guarantees on predictive mean, variance, and negative log-likelihood. Our analysis shows how the quality of the approximation depends on the kernel spectrum and sketch size, and our experiments demonstrate consistent improvements over existing baselines.

## Broader Impact Statement

We do not foresee any potential negative impact.

## Acknowledgments

Anirban Dasgupta would like to acknowledge the following grants and the corresponding funding agencies—SERB-MATRICS grant, SERB-CRG research grant.

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

## A    Appendix

In this section, we present our main theoretical results for the predictive mean (Theorem 5), variance (Theorem 7), and negative log-likelihood (Theorem 8) under ridge leverage score–based sketching. We begin by outlining the setup, notation, and assumptions common to all three theorems.

### A.1    Common Theoretical Setup:

Let $\mathcal{D} = \{(x_i, y_i)\}_{i=1}^{n}$ be the dataset, where $x_i \in \mathbb{R}^d$ are the input points and $y_i \in \mathbb{R}$ are the corresponding outputs. Let $k(x, x')$ be the kernel function used in the Gaussian process regression, and let $K$ be the kernel matrix $n \times n$ such that $K_{ij} = k(x_i, x_j)$. Let the eigenvalue decomposition of the kernel matrix, $K = U\Sigma U^T$.

Let $S \in \mathbb{R}^{n \times m}$ be a sketching matrix (obtained in Algorithm 1) so that $S_{ij} = \sqrt{\frac{1}{mp_i}}$ if $i = i_j$ else 0, where $m \ll n$ is obtained by probability distribution $(p_i)_{1 \leq i \leq n}$ such that $\forall i \in \{1, \cdots, n\}, \quad p_i \geq \beta \cdot l_i(\sigma_\xi^2)/\sum_{i=1}^{n} l_i(\sigma_\xi^2)$ for some $\beta \in (0, 1]$.

We define the sketch of a kernel matrix $L_\gamma = KS(S^\top KS + \gamma I)^{-1}S^\top K$ as the submatrix of $K$ where $\gamma > 0$.

Moreover, let

$$D = \Phi - \Phi^{1/2}U^\top SS^\top U\Phi^{1/2}$$

with $\Phi = \Sigma(\Sigma + \gamma I)^{-1}$.

From, El Alaoui & Mahoney (2015) we have that, as long as the sketching matrix $S$ satisfies $\lambda_{max}(D) \leq t$ for $t \in (0, 1)$ and $\lambda_{max}$ denoting the maximum eigenvalue. Under this condition, they show that the approximation error between the original kernel matrix $K$ and its sketched version $L_\gamma$ is bounded as

$$0 \preceq K - L_\gamma \preceq \left(\frac{\gamma}{1-t}\right)I$$

To ensure this bound holds, we follow the **sketch size guarantee** provided in Theorem 2 of Appendix B in El Alaoui & Mahoney (2015), which characterizes the **required number of samples** $m$ based on the ridge leverage score distribution is,

$$m \geq 8 \left(\frac{d_{\text{eff}}}{\beta} + \frac{1}{6}\right) \log\left(\frac{n}{\delta}\right)$$

For the mean and variance inference on test data, we express the test kernel vector as $k_* = U\alpha$, where $\alpha \in \mathbb{R}^n$ with $\alpha_i = u_i^\top k_*$, and $U = [u_1, \ldots, u_n]$ is the eigenvector matrix of $K$. This is a standard change-of-basis in the eigenspace of $K$ and does not introduce any additional modeling assumptions.

**Interpretation of $\alpha$ and $\beta$.** The coefficients $\alpha$ quantify the alignment of the test point with the spectral components of the kernel matrix, and determine how the eigenvalue spectrum influences the approximation error.

The parameter $\beta \in (0, 1]$ controls the quality of the sampling distribution relative to the ridge leverage scores through the condition $p_i \geq \beta \ell_i / \sum_j \ell_j$. Larger $\beta$ corresponds to more accurate leverage-based sampling (e.g., exact leverage scores when $\beta = 1$), while smaller $\beta$ allows approximate scores and leads to a larger required sketch size $m = \tilde{O}(d_{\text{eff}}/\beta)$.

## B    Predictive Mean Estimation using Sketching

**Theorem 5** (Predictive Mean Approximation under $\sigma_\xi^2-$ Ridge Leverage Score Sketching). *For the notations and assumptions defined in our **Setup** let*

$$\mu(x^*) = k_*^T(K + \sigma_\xi^2 I)^{-1}y$$

*be the predictive mean of Gaussian Process Regression at a new point $x^*$ for the full kernel matrix. Here $k_* = [k(x_1, x^*), k(x_2, x^*), \ldots, k(x_n, x^*)]^T$ and $\sigma_\xi^2$ is the noise. For the sketch of a kernel matrix $L_\gamma$, the predictive mean is,*

$$\mu_S(x^*) = k_*^T (L_\gamma + \sigma_\xi^2 I)^{-1} y$$

*For the $L_\gamma$ obtained using Algorithm 1 we have*

$$|\mu(x^*) - \mu_S(x^*)| \leq \left(\frac{\gamma}{1-t}\right) \sqrt{\sum_{i=1}^{n} \frac{\alpha_i^2}{(\Sigma_{i,i} + \sigma_\xi^2)^2}} \cdot \|y\|_2 \cdot \lambda_{max}(\Delta_D)$$

*where, $\Delta_D = \left(\Sigma \left[I - \frac{\gamma}{1-t}(\Sigma + \gamma I)^{-1}\right] + \sigma_\xi^2 I\right)^{-1}$, holds with probability at least $1 - \delta$, if the sketch size $m$ is set so that*

$$m \geq 8 \left(\frac{d_{eff}}{\beta} + \frac{1}{6}\right) \log\left(\frac{n}{\delta}\right)$$

*Proof.*

$$\begin{aligned}
|\mu(x^*) - \mu_S(x^*)| &= \left|k_*^T (K + \sigma_\xi^2 I)^{-1} y - k_*^T (L_\gamma + \sigma_\xi^2 I)^{-1} y\right| \\
&= \left|k_*^T [(K + \sigma_\xi^2 I)^{-1} - (L_\gamma + \sigma_\xi^2 I)^{-1}] y\right| \\
&\leq \left\|k_*^T [(K + \sigma_\xi^2 I)^{-1} - (L_\gamma + \sigma_\xi^2 I)^{-1}]\right\|_2 \cdot \|y\|_2 \\
&= \left\|k_*^T [(K + \sigma_\xi^2 I)^{-1} (L_\gamma - K)(L_\gamma + \sigma_\xi^2 I)^{-1}]\right\|_2 \cdot \|y\|_2 \\
&= \left\|k_*^T [(K + \sigma_\xi^2 I)^{-1} (K - L_\gamma)(L_\gamma + \sigma_\xi^2 I)^{-1}]\right\|_2 \cdot \|y\|_2 \\
&\leq \left\|k_*^T [(K + \sigma_\xi^2 I)^{-1} (K - L_\gamma)]\right\|_2 \left\|(L_\gamma + \sigma_\xi^2 I)^{-1}\right\|_{op} \cdot \|y\|_2 \\
&\leq \left(\frac{\gamma}{1-t}\right) \left\|k_*^T (K + \sigma_\xi^2 I)^{-1}\right\|_2 \left\|(L_\gamma + \sigma_\xi^2 I)^{-1}\right\|_{op} \cdot \|y\|_2 \quad \left(\text{As, } 0 \preceq K - L_\gamma \preceq \left(\frac{\gamma}{1-t}\right) I\right) \\
&\leq \left(\frac{\gamma}{1-t}\right) \left\|k_*^T (K + \sigma_\xi^2 I)^{-1}\right\|_2 \cdot \|y\|_2 \cdot \lambda_{max}(\Delta_D) \\
&\qquad \left(\text{Putting bound on, } \left\|(L_\gamma + \sigma_\xi^2 I)^{-1}\right\|_{op} \text{ from equation } 22 \text{ below}\right) \\
&= \left(\frac{\gamma}{1-t}\right) \left\|k_*^T (U \Sigma U^\top + \sigma_\xi^2 I)^{-1}\right\|_2 \cdot \|y\|_2 \cdot \lambda_{max}(\Delta_D) \\
&= \left(\frac{\gamma}{1-t}\right) \left\|k_*^\top (U(\Sigma + \sigma_\xi^2 I) U^\top)^{-1}\right\|_2 \cdot \|y\|_2 \cdot \lambda_{max}(\Delta_D) \\
&= \left(\frac{\gamma}{1-t}\right) \left\|k_*^\top (U(\Sigma + \sigma_\xi^2 I)^{-1} U^\top)\right\|_2 \cdot \|y\|_2 \cdot \lambda_{max}(\Delta_D) \\
&= \left(\frac{\gamma}{1-t}\right) \left\|(U\alpha)^\top (U(\Sigma + \sigma_\xi^2 I)^{-1} U^\top)\right\|_2 \cdot \|y\|_2 \cdot \lambda_{max}(\Delta_D) \quad (\text{Substitute, } k_* = U\alpha) \\
&= \left(\frac{\gamma}{1-t}\right) \left\|\alpha^\top U^\top U(\Sigma + \sigma_\xi^2 I)^{-1} U^\top\right\|_2 \cdot \|y\|_2 \cdot \lambda_{max}(\Delta_D) \\
&= \left(\frac{\gamma}{1-t}\right) \left\|\alpha^\top (\Sigma + \sigma_\xi^2 I)^{-1}\right\|_2 \cdot \|y\|_2 \cdot \lambda_{max}(\Delta_D)
\end{aligned}$$

The spectral norm decomposes as:

$$\left\|\alpha^T (\Sigma + \sigma_\xi^2 I)^{-1}\right\|_2 = \sqrt{\sum_{i=1}^{n} \alpha_i^2 \left(\frac{1}{(\Sigma_{i,i} + \sigma_\xi^2)}\right)^2}$$

Therefore, in the final bound we have

$$|\mu(x^*) - \mu_S(x^*)| \leq \left(\frac{\gamma}{1-t}\right) \sqrt{\sum_{i=1}^{n} \frac{\alpha_i^2}{(\Sigma_{i,i} + \sigma_\xi^2)^2}} \cdot \|y\|_2 \cdot \lambda_{max}(\Delta_D)$$

$\square$

**Corollary 5** (Predictive Mean Approximation under Polynomial Spectrum and Source Condition). *Let $K = U\Sigma U^\top$ with eigenvalues satisfying polynomial decay*

$$\sigma_i \asymp i^{-\rho}, \qquad \rho > 1,$$

*and assume the test kernel vector satisfies the source condition of order $r > 1/2$:*

$$k_* = U\alpha, \qquad \alpha_i^2 \leq C\,\sigma_i^{1+2r}, \quad i = 1,\dots,n.$$

*Assume the Nystrom sketch satisfies the ridge leverage condition with*

$$m \gtrsim d_{\text{eff}}(\sigma_\xi^2)\log(n/\delta), \qquad d_{\text{eff}}(\lambda) = \text{Tr}\big(K(K+\lambda I)^{-1}\big),$$

*and that $\gamma \asymp \sigma_\xi^2$.*

*Let $\sigma_\xi^2 \asymp n^{-\beta}$ for $\beta > 0$. Then with probability at least $1 - \delta$,*

$$|\mu(x_*) - \mu_S(x_*)| = O\Big(\sigma_y\, n^{-\beta\left(r-\frac{1}{2}-\frac{1}{2\rho}\right)}\Big), \qquad m = O\Big(n^{\beta/\rho}\log n\Big),$$

*where $\sigma_y^2 = \mathbb{E}[\|y\|_2^2]$ is the expected squared norm of the observation vector.*

*In particular, for the standard scaling $\sigma_\xi^2 \asymp n^{-1}$ (i.e., $\beta = 1$), this yields*

$$|\mu(x_*) - \mu_S(x_*)| = O\Big(\sigma_y\, n^{-\left(r-\frac{1}{2}-\frac{1}{2\rho}\right)}\Big), \qquad m = O\Big(n^{1/\rho}\log n\Big)$$

*Proof.* From Theorem 5, we have

$$|\mu(x_*) - \mu_S(x_*)| \leq \frac{\gamma}{1-t}\|y\|_2 \sqrt{\sum_{i=1}^{n} \frac{\alpha_i^2}{(\sigma_i + \sigma_\xi^2)^2}} \cdot \lambda_{\max}(\Delta_D)$$

**Step 1: Apply Nystrom bounds.** Using the ridge leverage Nystrom bounds,

$$\lambda_{\max}(\Delta_D) = O\big((\sigma_\xi^2)^{-1}\big), \qquad \gamma \asymp \sigma_\xi^2,$$

and $t \in (0,1)$ (which holds under the stated sketch size condition), we obtain

$$|\mu(x_*) - \mu_S(x_*)| = O\left(\|y\|_2 \sqrt{\sum_{i=1}^{n} \frac{\alpha_i^2}{(\sigma_i + \sigma_\xi^2)^2}}\right)$$

**Step 2: Apply source condition and bound the sum.** By the source condition $\alpha_i^2 \leq C\,\sigma_i^{1+2r}$,

$$S = \sum_{i=1}^{n} \frac{\alpha_i^2}{(\sigma_i + \sigma_\xi^2)^2} \leq C\sum_{i=1}^{n} \frac{\sigma_i^{1+2r}}{(\sigma_i + \sigma_\xi^2)^2}$$

Let $i^*$ be the index satisfying $\sigma_{i^*} \asymp \sigma_\xi^2$. Since $\sigma_i \asymp i^{-\rho}$, we have

$$i^* \asymp (\sigma_\xi^2)^{-1/\rho}$$

Splitting the sum at $i^*$:

$$S \le C \left( \sum_{i \le i^*} \frac{\sigma_i^{1+2r}}{(\sigma_i + \sigma_\xi^2)^2} + \sum_{i > i^*} \frac{\sigma_i^{1+2r}}{(\sigma_i + \sigma_\xi^2)^2} \right)$$

For $i \le i^*$: We have $\sigma_i \gtrsim \sigma_\xi^2$, so $\sigma_i + \sigma_\xi^2 \asymp \sigma_i$. Thus,

$$\sum_{i \le i^*} \frac{\sigma_i^{1+2r}}{(\sigma_i + \sigma_\xi^2)^2} \asymp \sum_{i \le i^*} \sigma_i^{2r-1} \asymp \sum_{i \le i^*} i^{-\rho(2r-1)}$$

Since $r > 1/2$, we have $\rho(2r-1) > 0$. For $\rho(2r-1) < 1$, the sum is dominated by the largest term at $i = i^*$:

$$\sum_{i \le i^*} i^{-\rho(2r-1)} \asymp (i^*)^{1-\rho(2r-1)} \asymp (\sigma_\xi^2)^{-(1-\rho(2r-1))/\rho} = (\sigma_\xi^2)^{2r-1-1/\rho}$$

For $\rho(2r-1) \ge 1$, the sum scales as

$$\sum_{i \le i^*} i^{-\rho(2r-1)} \asymp (i^*)^{-\rho(2r-1)} \asymp (\sigma_\xi^2)^{2r-1-1/\rho}$$

For $i > i^*$: We have $\sigma_i \ll \sigma_\xi^2$, so $\sigma_i + \sigma_\xi^2 \asymp \sigma_\xi^2$. Thus,

$$\sum_{i > i^*} \frac{\sigma_i^{1+2r}}{(\sigma_\xi^2)^2} \asymp (\sigma_\xi^2)^{-2} \sum_{i > i^*} i^{-\rho(1+2r)} \asymp (\sigma_\xi^2)^{-2} (i^*)^{1-\rho(1+2r)},$$

where the last step uses $\rho(1 + 2r) > \rho > 1$. This gives

$$(\sigma_\xi^2)^{-2}(\sigma_\xi^2)^{-(1-\rho(1+2r))/\rho} = (\sigma_\xi^2)^{-2-1/\rho+1+2r} = (\sigma_\xi^2)^{2r-1-1/\rho}$$

Both sums yield the same scaling, so

$$S \asymp (\sigma_\xi^2)^{2r-1-1/\rho}$$

Therefore,

$$|\mu(x_*) - \mu_S(x_*)| = O\left( \|y\|_2 (\sigma_\xi^2)^{r-\frac{1}{2}-\frac{1}{2\rho}} \right)$$

**Step 3: Express in terms of sample size $n$.** The effective dimension satisfies

$$d_{\mathrm{eff}}(\sigma_\xi^2) = \sum_{i=1}^{n} \frac{\sigma_i}{\sigma_i + \sigma_\xi^2} \asymp i^* \asymp (\sigma_\xi^2)^{-1/\rho}$$

Setting $\sigma_\xi^2 \asymp n^{-\beta}$, we obtain

$$|\mu(x_*) - \mu_S(x_*)| = O\left( \|y\|_2 \, n^{-\beta\left(r-\frac{1}{2}-\frac{1}{2\rho}\right)} \right),$$

and the sketch size requirement becomes

$$m = O\big(d_{\mathrm{eff}}(\sigma_\xi^2) \log n\big) = O\left(n^{\beta/\rho} \log n\right)$$

For the standard choice $\beta = 1$ (i.e., $\sigma_\xi^2 \asymp n^{-1}$), the rates simplify as stated. $\qquad \square$

**Lemma 6** (Spectral lower bound). *For any $t \in (0,1)$, $\gamma > 0$ and all $\sigma \ge 0$,*

$$\sigma - \frac{\gamma\sigma}{(1-t)(\sigma+\gamma)} \ge -\frac{t\gamma}{1-t+t}$$

*Proof.* Define

$$f(\sigma) = \sigma - \frac{\gamma\sigma}{(1-t)(\sigma+\gamma)}, \qquad \sigma \geq 0$$

Then

$$f'(\sigma) = 1 - \frac{\gamma^2}{(1-t)(\sigma+\gamma)^2}$$

Setting $f'(\sigma) = 0$ gives the unique minimizer

$$\sigma^\star = \frac{\gamma}{\sqrt{1-t}} - \gamma$$

Evaluating $f(\sigma^\star)$ yields

$$f(\sigma^\star) = -\frac{t\gamma}{1-t+t}$$

Since $f(\sigma) \to 0$ as $\sigma \to \infty$, this is the global minimum. $\qquad\square$

**Corollary 6** (Uniform Predictive Mean Approximation)**.** *Assume that the kernel is uniformly bounded on a compact domain $\mathcal{X}$, i.e.,*

$$\sup_{x\in\mathcal{X}} k(x,x) \leq \kappa^2$$

*and that the sketching parameters satisfy*

$$\sigma_\xi^2 > \frac{t\gamma}{1-t+t}$$

*Then, with probability at least $1-\delta$ over the sketching randomness, the predictive mean error satisfies*

$$\sup_{x^*\in\mathcal{X}} |\mu(x^*) - \mu_S(x^*)| \leq \frac{\gamma\kappa}{(1-t)\sqrt{\sigma_\xi^2}} \cdot \frac{\|y\|_2}{\sigma_\xi^2 - \frac{t\gamma}{1-t+t}}$$

*Proof.* From Theorem 5, for any fixed test point $x^* \in \mathcal{X}$,

$$|\mu(x^*) - \mu_S(x^*)| \leq \frac{\gamma}{1-t} \cdot \lambda_{\max}(\Delta_D) \cdot \sqrt{\sum_{i=1}^n \frac{\alpha_i^2}{(\Sigma_{ii} + \sigma_\xi^2)^2}} \cdot \|y\|_2 \tag{14}$$

where $\alpha = U^\top k_*$ and $k_* = (k(x_1,x^*),\ldots,k(x_n,x^*))^\top$.

**Step 1: Bounding the spectral alignment term.** We first rewrite the spectral term using the eigendecomposition $K = U\Sigma U^\top$:

$$\sum_{i=1}^n \frac{\alpha_i^2}{(\Sigma_{ii} + \sigma_\xi^2)^2} = \sum_{i=1}^n \frac{(U^\top k_*)_i^2}{(\sigma_i + \sigma_\xi^2)^2}$$
$$= k_*^\top U(\Sigma + \sigma_\xi^2 I)^{-2} U^\top k_*$$
$$= k_*^\top (K + \sigma_\xi^2 I)^{-2} k_* \tag{15}$$

To bound this quantity uniformly over $\mathcal{X}$, we exploit the eigenvalue structure. Since $\sigma_i + \sigma_\xi^2 \geq \sigma_\xi^2$ for all $i \in \{1,\ldots,n\}$, we have

$$\frac{1}{(\sigma_i + \sigma_\xi^2)^2} \leq \frac{1}{\sigma_\xi^2(\sigma_i + \sigma_\xi^2)}$$

Therefore,

$$
\begin{aligned}
k_*^\top (K + \sigma_\xi^2 I)^{-2} k_* &= \sum_{i=1}^n \frac{\alpha_i^2}{(\sigma_i + \sigma_\xi^2)^2} \\
&\leq \frac{1}{\sigma_\xi^2} \sum_{i=1}^n \frac{\alpha_i^2}{\sigma_i + \sigma_\xi^2} \\
&= \frac{1}{\sigma_\xi^2} k_*^\top (K + \sigma_\xi^2 I)^{-1} k_*
\end{aligned}
\tag{16}
$$

From the Gaussian Process posterior variance formula, we have

$$
\mathrm{Var}(f(x^*) \mid \mathcal{D}) = k(x^*, x^*) - k_*^\top (K + \sigma_\xi^2 I)^{-1} k_* \geq 0
$$

which implies

$$
k_*^\top (K + \sigma_\xi^2 I)^{-1} k_* \leq k(x^*, x^*) \leq \kappa^2
\tag{17}
$$

Combining equation 16 and equation 17, we obtain

$$
\sup_{x^* \in \mathcal{X}} \sqrt{\sum_{i=1}^n \frac{\alpha_i^2}{(\Sigma_{ii} + \sigma_\xi^2)^2}} \leq \frac{\kappa}{\sqrt{\sigma_\xi^2}}
\tag{18}
$$

**Step 2: Bounding the maximum eigenvalue of $\Delta_D$.** Recall that

$$
\Delta_D = \left( \Sigma \left[ I - \frac{\gamma}{1-t} (\Sigma + \gamma I)^{-1} \right] + \sigma_\xi^2 I \right)^{-1}
$$

The $i$-th eigenvalue of $\Delta_D^{-1}$ is given by

$$
\lambda_i^{(\mathrm{inner})} = \sigma_i \left[ 1 - \frac{\gamma}{(1-t)(\sigma_i + \gamma)} \right] + \sigma_\xi^2 = \sigma_i - \frac{\gamma \sigma_i}{(1-t)(\sigma_i + \gamma)} + \sigma_\xi^2
\tag{19}
$$

To find $\lambda_{\max}(\Delta_D)$, we must determine $\min_i \lambda_i^{(\mathrm{inner})}$.

*Case 1: Small eigenvalues.* For $\sigma_i \ll \gamma$, we can approximate

$$
\frac{\gamma \sigma_i}{(1-t)(\sigma_i + \gamma)} \approx \frac{\gamma \sigma_i}{(1-t)\gamma} = \frac{\sigma_i}{1-t}
$$

so that

$$
\lambda_i^{(\mathrm{inner})} \approx \sigma_i - \frac{\sigma_i}{1-t} + \sigma_\xi^2 = \sigma_i \left( 1 - \frac{1}{1-t} \right) + \sigma_\xi^2 = -\frac{t \sigma_i}{1-t} + \sigma_\xi^2
$$

For small $\sigma_i$, this approaches $\sigma_\xi^2$ from below.

*Case 2: Large eigenvalues.* For $\sigma_i \gg \gamma$, we have

$$
\frac{\gamma \sigma_i}{(1-t)(\sigma_i + \gamma)} \approx \frac{\gamma \sigma_i}{(1-t)\sigma_i} = \frac{\gamma}{1-t}
$$

so that

$$
\lambda_i^{(\mathrm{inner})} \approx \sigma_i - \frac{\gamma}{1-t} + \sigma_\xi^2
$$

As $\sigma_i \to \infty$, this grows without bound.

*Critical observation.* Rewriting equation 19 as

$$\lambda_i^{(\text{inner})} = \frac{\sigma_i[(1-t)\sigma_i - t\gamma]}{(1-t)(\sigma_i+\gamma)} + \sigma_\xi^2$$

we see that the first term is negative only when

$$0 < \sigma_i < \frac{t\gamma}{1-t}$$

Over this interval the function

$$g(\sigma_i) = \frac{\gamma\sigma_i}{(1-t)(\sigma_i+\gamma)}$$

is increasing in $\sigma_i$, hence its maximum is attained at

$$\sigma_i = \frac{t\gamma}{1-t}$$

Substituting this value yields

$$\max_{0<\sigma_i<\frac{t\gamma}{1-t}} g(\sigma_i) = \frac{t\gamma}{1-t+t}$$

Therefore by above and Lemma 6,

$$\min_i \lambda_i^{(\text{inner})} \geq \sigma_\xi^2 - \frac{t\gamma}{1-t+t}$$

$$\lambda_{\max}(\Delta_D) \leq \frac{1}{\sigma_\xi^2 - \frac{t\gamma}{1-t+t}} \tag{20}$$

**Step 3: Combining the bounds.** Substituting equation 18 and equation 20 into equation 14, we obtain

$$\sup_{x^*\in\mathcal{X}} |\mu(x^*) - \mu_S(x^*)| \leq \frac{\gamma}{1-t} \cdot \frac{1}{\sigma_\xi^2 - \frac{t\gamma}{1-t+t}} \cdot \frac{\kappa}{\sqrt{\sigma_\xi^2}} \cdot \|y\|_2$$

which simplifies to the stated bound. This completes the proof.

$$\square$$

## C  Predictive Variance Estimation using Sketching

**Theorem 7** (Predictive Variance Approximation under $\sigma_\xi^2-$ Ridge Leverage Score Sketching). *For the notations and assumptions defined in our **Setup** let*

$$Var(x^*) = k(x^*,x^*) - k_*^\top(K + \sigma_\xi^2 I)^{-1}k_*$$

*be the predictive variance of Gaussian Process Regression at a new point $x^*$. Here, $k_* = [k(x_1,x^*), k(x_2,x^*), \ldots, k(x_n,x^*)]^T$ and $\sigma_\xi^2$ is the noise variance. For the sketch of a kernel matrix $L_\gamma$, the predictive variance is,*

$$Var_S(x^*) = k(x^*,x^*) - k_*^\top(L_\gamma + \sigma_\xi^2 I)^{-1}k_*$$

*For the $L_\gamma$ obtained using Algorithm 1 we have*

$$|Var(x^*) - Var_S(x^*)| \leq \left(\frac{\gamma}{1-t}\right)\|\alpha^\top\Delta_D\|_2 \cdot \sqrt{\sum_{i=1}^n \alpha_i^2\left(\frac{1}{(\Sigma_{i,i}+\sigma_\xi^2)}\right)^2}$$

*where, $\Delta_D = \left(\Sigma\left[I - \frac{\gamma}{1-t}(\Sigma+\gamma I)^{-1}\right] + \sigma_\xi^2 I\right)^{-1}$, holds with probability at least $1-\delta$ if the sketch size $m$ is set so that*

$$m \geq 8\left(\frac{d_{eff}}{\beta} + \frac{1}{6}\right)\log\left(\frac{n}{\delta}\right)$$

*Proof.*

$$
\begin{aligned}
|Var(x^*) - Var_S(x^*)| &= \left| k(x^*, x^*) - k_*^T(K + \sigma_\xi^2 I)^{-1}k_* - k(x^*, x^*) + k_*^T(L_\gamma + \sigma_\xi^2 I)^{-1}k_* \right| \\
&= \left| -k_*^T(K + \sigma_\xi^2 I)^{-1}k_* + k_*^T(L_\gamma + \sigma_\xi^2 I)^{-1}k_* \right| \\
&= \left| k_*^T(L_\gamma + \sigma_\xi^2 I)^{-1}k_* - k_*^T(K + \sigma_\xi^2 I)^{-1}k_* \right| \\
&= \left| k_*^T[(L_\gamma + \sigma_\xi^2 I)^{-1} - (K + \sigma_\xi^2 I)^{-1}]k_* \right| \\
&= \left| k_*^T[(L_\gamma + \sigma_\xi^2 I)^{-1}(K - L_\gamma)(K + \sigma_\xi^2 I)^{-1}]k_* \right| \\
&\leq \left( \frac{\gamma}{1-t} \right) \left| k_*^T[(L_\gamma + \sigma_\xi^2 I)^{-1}(K + \sigma_\xi^2 I)^{-1}]k_* \right| &&\left(\text{As, } 0 \preceq K - L_\gamma \preceq \left( \tfrac{\gamma}{1-t} \right) I\right) \\
&= \left( \frac{\gamma}{1-t} \right) \left| k_*^T[(L_\gamma + \sigma_\xi^2 I)^{-1}(U\Sigma U^\top + \sigma_\xi^2 I)^{-1}]k_* \right| \\
&= \left( \frac{\gamma}{1-t} \right) \left| k_*^T[(L_\gamma + \sigma_\xi^2 I)^{-1}(U(\Sigma + \sigma_\xi^2 I)U^\top)^{-1}]k_* \right| \\
&= \left( \frac{\gamma}{1-t} \right) \left| (U\alpha)^T[(L_\gamma + \sigma_\xi^2 I)^{-1}(U(\Sigma + \sigma_\xi^2 I)^{-1}U^\top)](U\alpha) \right| &&(\text{Substitute, } k_* = U\alpha) \\
&\leq \left( \frac{\gamma}{1-t} \right) \left| \alpha^\top U^\top(L'_\gamma + \sigma_\xi^2 I)^{-1}(U(\Sigma + \sigma_\xi^2 I)^{-1}U^\top)U\alpha \right| &&(\text{Since, } L_\gamma \succeq L'_\gamma) \\
&= \left( \frac{\gamma}{1-t} \right) \left| \alpha^\top U^\top \left( U\left[ \Sigma - \left( \frac{\gamma}{1-t} \right)\Sigma(\Sigma + \gamma I)^{-1} \right]U^T + \sigma_\xi^2 I \right)^{-1}(U(\Sigma + \sigma_\xi^2 I)^{-1}U^\top)U\alpha \right| \\
&= \left( \frac{\gamma}{1-t} \right) \left| \alpha^\top U^\top \left( U\left[ \Sigma - \left( \frac{\gamma}{1-t} \right)\Sigma(\Sigma + \gamma I)^{-1} + \sigma_\xi^2 I \right]U^T \right)^{-1}(U(\Sigma + \sigma_\xi^2 I)^{-1}U^\top)U\alpha \right| \\
&= \left( \frac{\gamma}{1-t} \right) \left| \alpha^\top \left[ \Sigma - \left( \frac{\gamma}{1-t} \right)\Sigma(\Sigma + \gamma I)^{-1} + \sigma_\xi^2 I \right]^{-1}(\Sigma + \sigma_\xi^2 I)^{-1}\alpha \right| \\
&= \left( \frac{\gamma}{1-t} \right) \left| \alpha^\top \Delta_D (\Sigma + \sigma_\xi^2 I)^{-1}\alpha \right| &&\left(\text{As, } \Delta_D = \left[ \Sigma - \left( \tfrac{\gamma}{1-t} \right)\Sigma(\Sigma + \gamma I)^{-1} + \sigma_\xi^2 I \right]^{-1}\right) \\
&\leq \left( \frac{\gamma}{1-t} \right) \left\| \alpha^\top \Delta_D \right\|_2 \left\| (\Sigma + \sigma_\xi^2 I)^{-1}\alpha \right\|_2
\end{aligned}
$$

The spectral norm decomposes as:

$$
\left\| (\Sigma + \sigma_\xi^2 I)^{-1}\alpha \right\|_2 = \sqrt{\sum_{i=1}^n \alpha_i^2 \left( \frac{1}{(\Sigma_{i,i} + \sigma_\xi^2)} \right)^2}
$$

Therefore, in the final bound we have

$$
|Var(x^*) - Var_S(x^*)| \leq \left( \frac{\gamma}{1-t} \right) \left\| \alpha^\top \Delta_D \right\|_2 \cdot \sqrt{\sum_{i=1}^n \alpha_i^2 \left( \frac{1}{(\Sigma_{i,i} + \sigma_\xi^2)} \right)^2}
$$

$\square$

**Corollary 7** (Uniform Predictive Variance Approximation). *Assume that the kernel is uniformly bounded on a compact domain $\mathcal{X}$, i.e.,*

$$
\sup_{x \in \mathcal{X}} k(x, x) \leq \kappa^2
$$

*and that the sketching parameters satisfy*

$$\sigma_\xi^2 > \frac{t\gamma}{1 - t + t}$$

*Then, with probability at least $1 - \delta$ over the sketching randomness, the predictive variance error satisfies*

$$\sup_{x^* \in \mathcal{X}} \left| \mathrm{Var}(x^*) - \mathrm{Var}_S(x^*) \right| \leq \frac{\gamma \, \kappa^3 \sqrt{n}}{(1 - t)\sqrt{\sigma_\xi^2}} \cdot \frac{1}{\sigma_\xi^2 - \frac{t\gamma}{1-t+t}}$$

*Proof.* From Theorem 7, for any fixed $x^* \in \mathcal{X}$,

$$\left| \mathrm{Var}(x^*) - \mathrm{Var}_S(x^*) \right| \leq \frac{\gamma}{1 - t} \cdot \|\alpha^\top \Delta_D\|_2 \cdot \left\| (\Sigma + \sigma_\xi^2 I)^{-1} \alpha \right\|_2 \tag{21}$$

**Step 1: Bounding the spectral alignment term.** Using $K = U\Sigma U^\top$,

$$\left\| (\Sigma + \sigma_\xi^2 I)^{-1} \alpha \right\|_2^2 = k_*^\top (K + \sigma_\xi^2 I)^{-2} k_*$$

Since $(\sigma_i + \sigma_\xi^2)^{-2} \leq \sigma_\xi^{-2} (\sigma_i + \sigma_\xi^2)^{-1}$,

$$k_*^\top (K + \sigma_\xi^2 I)^{-2} k_* \leq \frac{1}{\sigma_\xi^2} k_*^\top (K + \sigma_\xi^2 I)^{-1} k_*$$

By the GP posterior variance identity,

$$k_*^\top (K + \sigma_\xi^2 I)^{-1} k_* \leq k(x^*, x^*) \leq \kappa^2$$

hence

$$\sup_{x^* \in \mathcal{X}} \left\| (\Sigma + \sigma_\xi^2 I)^{-1} \alpha \right\|_2 \leq \frac{\kappa}{\sqrt{\sigma_\xi^2}}$$

**Step 2: Bounding $\|\alpha^\top \Delta_D\|_2$.** Since $\Delta_D$ is diagonal in the eigenbasis,

$$\|\alpha^\top \Delta_D\|_2 \leq \lambda_{\max}(\Delta_D) \, \|\alpha\|_2$$

From the predictive mean corollary and Lemma 6,

$$\lambda_{\max}(\Delta_D) \leq \frac{1}{\sigma_\xi^2 - \frac{t\gamma}{1-t+t}}$$

Moreover,

$$\|\alpha\|_2 = \|k_*\|_2 = \left( \sum_{i=1}^{n} k(x_i, x^*)^2 \right)^{1/2} \leq \sqrt{n}\, \kappa^2$$

using $|k(x_i, x^*)| \leq \sqrt{k(x_i, x_i) k(x^*, x^*)} \leq \kappa^2$.
Hence,

$$\sup_{x^* \in \mathcal{X}} \|\alpha^\top \Delta_D\|_2 \leq \frac{\sqrt{n}\kappa^2}{\sigma_\xi^2 - \frac{t\gamma}{1-t+t}}$$

**Step 3: Combining the bounds.** Substituting into equation 21 gives

$$\sup_{x^* \in \mathcal{X}} \left| \mathrm{Var}(x^*) - \mathrm{Var}_S(x^*) \right| \leq \frac{\gamma}{1 - t} \cdot \frac{\sqrt{n}\kappa^2}{\sigma_\xi^2 - \frac{t\gamma}{1-t+t}} \cdot \frac{\kappa}{\sqrt{\sigma_\xi^2}}$$

which completes the proof. $\qquad \square$

## D   Negative Log Marginal Likelihood Approximation

**Theorem 8** (Negative Log Marginal Likelihood Approximation under $\sigma_\xi^2-$ Ridge Leverage Score Sketching)**.** *Let $K \in \mathbb{R}^{n \times n}$ be a symmetric positive semi-definite kernel matrix and $y \in \mathbb{R}^n$ the response vector. For $\sigma_\xi^2 > 0$, the negative log marginal likelihood (NLML) be,*

$$\mathcal{L}(K) = \frac{1}{2} y^\top (K + \sigma_\xi^2 I)^{-1} y + \frac{1}{2} \log \det(K + \sigma_\xi^2 I) + \frac{n}{2} \log(2\pi)$$

*The corresponding approximate NLML for the sketch of the kernel matrix $L_\gamma$ obtained using Algorithm 1 is given as,*

$$\mathcal{L}(L_\gamma) = \frac{1}{2} y^\top (L_\gamma + \sigma_\xi^2 I)^{-1} y + \frac{1}{2} \log \det(L_\gamma + \sigma_\xi^2 I) + \frac{n}{2} \log(2\pi)$$

*Then, for any $0 \le \delta \le 1$, if*

$$m \ge 8 \left( \frac{d_{eff}}{\beta} + \frac{1}{6} \right) \log \left( \frac{n}{\delta} \right)$$

*then, with probability at least $1 - \delta$ following inequality holds,*

$$|\mathcal{L}(K) - \mathcal{L}(L_\gamma)| \le \frac{\gamma}{2(1-t)} \left( \frac{1}{\lambda_{min}(K + \sigma_\xi^2 I)} \right) \cdot \|y\|_2^2 \cdot \lambda_{max}(\Delta_D) + \frac{\gamma}{2(1-t)} \mathrm{Tr}(\Delta_D)$$

*where, $\Delta_D = \left( \Sigma \left[ I - \frac{\gamma}{1-t} (\Sigma + \gamma I)^{-1} \right] + \sigma_\xi^2 I \right)^{-1}$.*

*Proof.* To analyze the negative log marginal likelihood (NLL), we omit the additive constant term involving $\frac{n}{2} \log(2\pi)$, as it does not affect the optimization or approximation. The resulting expression consists of two principal components: a quadratic term and a log-determinant term. We derive separate bounds for each of these components and then combine them to obtain an overall bound on the NLML approximation error.

$$\mathcal{L}(K) = \frac{1}{2} y^\top (K + \sigma_\xi^2 I)^{-1} y + \frac{1}{2} \log \det(K + \sigma_\xi^2 I) + \frac{n}{2} \log(2\pi)$$

Since, we have $0 \preceq K - L_\gamma \preceq \left( \frac{\gamma}{1-t} \right) I$ bound from El Alaoui & Mahoney (2015),

$$\frac{1}{2}\left|y^T(K+\sigma_\xi^2 I)^{-1}y - y^T(L_\gamma+\sigma_\xi^2 I)^{-1}y\right| = \frac{1}{2}\left|y^T[(K+\sigma_\xi^2 I)^{-1} - (L_\gamma+\sigma_\xi^2 I)^{-1}]y\right|$$

$$\leq \frac{1}{2}\left\|(K+\sigma_\xi^2 I)^{-1} - (L_\gamma+\sigma_\xi^2 I)^{-1}\right\|_{op} \cdot \|y\|_2^2$$

$$= \frac{1}{2}\left\|(K+\sigma_\xi^2 I)^{-1}(L_\gamma - K)(L_\gamma+\sigma_\xi^2 I)^{-1}\right\|_{op} \cdot \|y\|_2^2$$

$$= \frac{1}{2}\left\|(K+\sigma_\xi^2 I)^{-1}(K - L_\gamma)(L_\gamma+\sigma_\xi^2 I)^{-1}\right\|_{op} \cdot \|y\|_2^2$$

$$\leq \frac{1}{2}\left\|(K+\sigma_\xi^2 I)^{-1}(K - L_\gamma)\right\|_{op}\left\|(L_\gamma+\sigma_\xi^2 I)^{-1}\right\|_{op} \cdot \|y\|_2^2$$

$$\leq \frac{\gamma}{2(1-t)}\left\|(K+\sigma_\xi^2 I)^{-1}\right\|_{op}\left\|(L_\gamma+\sigma_\xi^2 I)^{-1}\right\|_{op} \cdot \|y\|_2^2$$

$$\left(\text{As, } 0 \preceq K - L_\gamma \preceq \left(\tfrac{\gamma}{1-t}\right)I\right)$$

$$\leq \frac{\gamma}{2(1-t)}\left\|(K+\sigma_\xi^2 I)^{-1}\right\|_{op} \cdot \|y\|_2^2 \cdot \lambda_{max}(\Delta_D)$$

$$\left(\text{Putting bound on, } \left\|(L_\gamma+\sigma_\xi^2 I)^{-1}\right\|_{op} \text{ from equation } 22 \text{ below}\right)$$

$$\leq \frac{\gamma}{2(1-t)}\left(\frac{1}{\lambda_{min}(K+\sigma_\xi^2 I)}\right) \cdot \|y\|_2^2 \cdot \lambda_{max}(\Delta_D)$$

For, $\left\|(L_\gamma+\sigma_\xi^2 I)^{-1}\right\|_{op}$ we can get upper bound like following,

$$L_\gamma = KS\left(S^T KS + \gamma I\right)^{-1}S^T K$$

With $K = U\Sigma U^\top$ and $R = \Sigma^{1/2}U^\top S$, $\bar{L}_\gamma = R(R^\top R + \gamma I)^{-1}R^\top$, we have

$$L_\gamma = U\Sigma^{1/2}\bar{L}_\gamma\Sigma^{1/2}U^\top$$

Due to the matrix inversion lemma, we have

$$\bar{L}_\gamma = RR^\top(RR^\top + \gamma I)^{-1}$$
$$= I - \gamma(RR^\top + \gamma I)^{-1}$$
$$= I - \gamma(\Sigma + \gamma I + RR^\top - \Sigma)^{-1}$$
$$= I - \gamma(\Sigma + \gamma I)^{-1/2}(I - D)^{-1}(\Sigma + \gamma I)^{-1/2}$$

with

$$D = (\Sigma + \gamma I)^{-1/2}(\Sigma - RR^\top)(\Sigma + \gamma I)^{-1/2}$$
$$= \Phi - \Phi^{1/2}U^\top SS^\top U\Phi^{1/2}$$

and $\Phi = \Sigma(\Sigma + \gamma I)^{-1}$.

If the sketching matrix $S$ satisfies $\lambda_{max}\left(\Phi - \Phi^{1/2}U^\top SS^\top U\Phi^{1/2}\right) = \lambda_{max}(D) \leq t$ for $t \in (0,1)$ where $\lambda_{max}$ denotes the maximum eigenvalue we can derive the lower bound for $\bar{L}_\gamma$ as the following,

$$\bar{L}_\gamma = I - \gamma(\Sigma + \gamma I)^{-1/2}(I - D)^{-1}(\Sigma + \gamma I)^{-1/2}$$

$$\succeq I - \gamma(\Sigma + \gamma I)^{-1/2}\left(\frac{1}{1-t}\right)I(\Sigma + \gamma I)^{-1/2} \qquad \text{(As, } (I - D)^{-1} \preceq \left(\frac{1}{1-t}\right)I)$$

$$= I - \left(\frac{\gamma}{1-t}\right)(\Sigma + \gamma I)^{-1}$$

Now, lets put this $\bar{L}_\gamma$ into $L_\gamma$,

$$L_\gamma = U\Sigma^{1/2}\bar{L}_\gamma\Sigma^{1/2}U^T$$

$$\succeq U\Sigma^{1/2}\left[I - \left(\frac{\gamma}{1-t}\right)(\Sigma + \gamma I)^{-1}\right]\Sigma^{1/2}U^T$$

$$= U\left[\Sigma - \left(\frac{\gamma}{1-t}\right)\Sigma(\Sigma + \gamma I)^{-1}\right]U^T$$

$$= L'_\gamma$$

Therefore, $L'_\gamma = U\left[\Sigma - \left(\frac{\gamma}{1-t}\right)\Sigma(\Sigma + \gamma I)^{-1}\right]U^T$ is the lower bound for $L_\gamma$.

For the upper bound of $\left\|(L_\gamma + \sigma_\xi^2 I)^{-1}\right\|_{op}$ term we need lower bound on $L_\gamma$ which is $L'_\gamma$,

$$\left\|(L_\gamma + \sigma_\xi^2 I)^{-1}\right\|_{op} \leq \left\|(L'_\gamma + \sigma_\xi^2 I)^{-1}\right\|_{op}$$

$$= \left\|\left(U\left[\Sigma - \left(\frac{\gamma}{1-t}\right)\Sigma(\Sigma + \gamma I)^{-1}\right]U^T + \sigma_\xi^2 I\right)^{-1}\right\|_{op}$$

$$= \left\|\left(U\left[\Sigma - \left(\frac{\gamma}{1-t}\right)\Sigma(\Sigma + \gamma I)^{-1} + \sigma_\xi^2 I\right]U^T\right)^{-1}\right\|_{op}$$

Using the eigendecomposition $K = U\Sigma U^\top$, and orthogonality of $U$, the operator norm simplifies as following,

Given the eigendecomposition of the kernel matrix $K = U\Sigma U^\top$, where $\Sigma = \text{diag}(\lambda_1, \ldots, \lambda_n)$, the following expression arises in the analysis,

$$\left\|\left(U\left[\Sigma - \left(\frac{\gamma}{1-t}\right)\Sigma(\Sigma + \gamma I)^{-1} + \sigma_\xi^2 I\right]U^\top\right)^{-1}\right\|_{op} = \left\|\left(\Sigma\left[I - \frac{\gamma}{1-t}(\Sigma + \gamma I)^{-1}\right] + \sigma_\xi^2 I\right)^{-1}\right\|_{op}$$

Lets denote, $\Delta_D = \left(\Sigma\left[I - \frac{\gamma}{1-t}(\Sigma + \gamma I)^{-1}\right] + \sigma_\xi^2 I\right)^{-1}$ then we have,

$$\left\|(L_\gamma + \sigma_\xi^2 I)^{-1}\right\|_{op} \leq \lambda_{max}(\Delta_D) \tag{22}$$

Now, lets bound log determinant term,

$$\frac{1}{2} \left| log \left| (K + \sigma_\xi^2 I) \right| - log \left| (L_\gamma + \sigma_\xi^2 I) \right| \right|$$

Now, we use proposition Bhatia (2013); Boyd & Vandenberghe (2004) which says for any two symmetric positive semi-definite matrix, $A \succ 0$ and $0 \preceq \Delta \preceq \alpha I$ following inequality holds,

$$\left| log \left| A + \Delta \right| - log \left| A \right| \right| \le Tr(A^{-1}\Delta) \le \alpha Tr(A^{-1})$$

In above proposition if, $A = L_\gamma + \sigma_\xi^2 I$ , $\Delta = K - L_\gamma$ and $\alpha = \left( \frac{\gamma}{1-t} \right)$ then,

$$
\begin{aligned}
\left| log \left| A + \Delta \right| - log \left| A \right| \right| &= \left| log \left| (L_\gamma + \sigma_\xi^2 I) + (K - L_\gamma) \right| - log \left| L_\gamma + \sigma_\xi^2 I \right| \right| \\
&= \left| log \left| (K + \sigma_\xi^2 I) \right| - log \left| (L_\gamma + \sigma_\xi^2 I) \right| \right| \\
&\le Tr \left[ (L_\gamma + \sigma_\xi^2 I)^{-1}(K - L_\gamma) \right] \\
&\le Tr \left[ (L_\gamma + \sigma_\xi^2 I)^{-1} \left( \frac{\gamma}{1-t} \right) I \right] \\
&= \left( \frac{\gamma}{1-t} \right) Tr \left[ (L_\gamma + \sigma_\xi^2 I)^{-1} \right] \\
&\le \left( \frac{\gamma}{1-t} \right) Tr \left[ (L_\gamma^{'} + \sigma_\xi^2 I)^{-1} \right] \qquad \text{(Since, } L_\gamma \succeq L_\gamma^{'} \text{ and } L_\gamma \text{ is PSD)} \\
&= \left( \frac{\gamma}{1-t} \right) Tr \left[ \left( U \left[ \Sigma - \left( \frac{\gamma}{1-t} \right) \Sigma(\Sigma + \gamma I)^{-1} \right] U^T + \sigma_\xi^2 I \right)^{-1} \right] \\
&= \left( \frac{\gamma}{1-t} \right) Tr \left[ \left( U \left[ \Sigma - \left( \frac{\gamma}{1-t} \right) \Sigma(\Sigma + \gamma I)^{-1} + \sigma_\xi^2 I \right] U^\top \right)^{-1} \right] \\
&= \left( \frac{\gamma}{1-t} \right) \mathrm{Tr} \left[ \left( \Sigma \left[ I - \frac{\gamma}{1-t}(\Sigma + \gamma I)^{-1} \right] + \sigma_\xi^2 I \right)^{-1} \right] \\
&\qquad\qquad\qquad\qquad\qquad\qquad\qquad\qquad\qquad \text{(Using the orthogonality of } U)
\end{aligned}
$$

Therefore bound for log determinant term is,

$$\frac{1}{2} \left| log \left| (K + \sigma_\xi^2 I) \right| - log \left| (L_\gamma + \sigma_\xi^2 I) \right| \right| \le \frac{\gamma}{2(1-t)} \mathrm{Tr} \left[ \left( \Sigma \left[ I - \frac{\gamma}{1-t}(\Sigma + \gamma I)^{-1} \right] + \sigma_\xi^2 I \right)^{-1} \right]$$

Since we are saying, $\Delta_D = \left( \Sigma \left[ I - \frac{\gamma}{1-t}(\Sigma + \gamma I)^{-1} \right] + \sigma_\xi^2 I \right)^{-1}$, we have,

$$\frac{1}{2} \left| log \left| (K + \sigma_\xi^2 I) \right| - log \left| (L_\gamma + \sigma_\xi^2 I) \right| \right| \le \frac{\gamma}{2(1-t)} \mathrm{Tr}(\Delta_D)$$

Now, combining both terms bound we will get overall bound,

$$\left| \mathcal{L}(K) - \mathcal{L}(L_\gamma) \right| \le \frac{\gamma}{2(1-t)} \left[ \left( \frac{1}{\lambda_{min}(K + \sigma_\xi^2 I)} \right) \cdot \|y\|_2^2 \cdot \lambda_{max}(\Delta_D) + \mathrm{Tr}(\Delta_D) \right]$$

$$\square$$

**Corollary 8** (Uniform NLML Approximation)**.** *Assume that the sketching parameters satisfy*

$$\sigma_\xi^2 > \frac{t\gamma}{1 - t + t}$$

*Then, with probability at least $1 - \delta$ over the sketching randomness, for all $y \in \mathbb{R}^n$,*

$$\left| \mathcal{L}(K) - \mathcal{L}(L_\gamma) \right| \leq \frac{\gamma}{2(1 - t)} \left[ \frac{\|y\|_2^2}{\sigma_\xi^2} \cdot \frac{1}{\sigma_\xi^2 - \frac{t\gamma}{1-t+t}} + \sum_{i=1}^n \frac{1}{\sigma_i - \frac{t\gamma}{1-t+t} + \sigma_\xi^2} \right]$$

*Proof.* From Theorem 8, with probability at least $1 - \delta$,

$$\left| \mathcal{L}(K) - \mathcal{L}(L_\gamma) \right| \leq \frac{\gamma}{2(1 - t)} \left[ \frac{\|y\|_2^2}{\lambda_{\min}(K + \sigma_\xi^2 I)} \cdot \lambda_{\max}(\Delta_D) + \mathrm{Tr}(\Delta_D) \right]$$

**Bounding the quadratic term.** Since $K \succeq 0$,

$$\lambda_{\min}(K + \sigma_\xi^2 I) \geq \sigma_\xi^2$$

From the spectral analysis of $\Delta_D$,

$$\lambda_{\max}(\Delta_D) \leq \frac{1}{\sigma_\xi^2 - \frac{t\gamma}{1-t+t}}$$

Therefore,

$$\frac{\|y\|_2^2}{\lambda_{\min}(K + \sigma_\xi^2 I)} \cdot \lambda_{\max}(\Delta_D) \leq \frac{\|y\|_2^2}{\sigma_\xi^2} \cdot \frac{1}{\sigma_\xi^2 - \frac{t\gamma}{1-t+t}}$$

**Bounding the log-determinant term.** By definition,

$$\mathrm{Tr}(\Delta_D) = \sum_{i=1}^n \frac{1}{\sigma_i - \frac{\gamma \sigma_i}{(1-t)(\sigma_i + \gamma)} + \sigma_\xi^2}$$

As shown in the predictive mean analysis,

$$\sigma_i - \frac{\gamma \sigma_i}{(1 - t)(\sigma_i + \gamma)} \geq -\frac{t\gamma}{1 - t + t}$$

hence

$$\mathrm{Tr}(\Delta_D) \leq \sum_{i=1}^n \frac{1}{\sigma_i - \frac{t\gamma}{1-t+t} + \sigma_\xi^2}$$

**Combining the bounds.** Substituting the above bounds yields the stated result. $\qquad\square$

# E   Additional Experimental Details

In this section, we provide supplementary experimental results obtained using the *Matern kernel* in addition to the RBF kernel used in the main paper. These additional experiments are designed to validate the robustness of our proposed Nystrom ridge leverage score sketching method across different kernel choices.

Consistent with our findings in the main text, the results with the Matern kernel confirm that our method continues to *outperform all baseline methods* across multiple datasets and evaluation metrics. In particular, it achieves *superior performance in terms of Negative Log Predictive Density (NLPD)*, a proper scoring rule that captures both prediction accuracy and uncertainty calibration. Our approach also yields *lower predictive mean and variance errors*, indicating more accurate posterior approximations of the underlying Gaussian Process Regression model.

These trends hold across all evaluated datasets, and the improvements remain significant despite the change in kernel. Importantly, the *approximate ridge leverage score algorithm* (used to efficiently construct the sketch matrix) remains effective, demonstrating both *scalability and predictive reliability* of our framework. Results from these additional experiments are presented in accompanying tables and figures, where the best-performing entries are shown in bold. We note that while several scalable variational methods exist, *SVGP* is included as a strong and widely adopted representative of the variational family. Our method demonstrates consistently lower *NLPD* and better uncertainty calibration than *SVGP*, *IterGP*, and other baselines across all kernel configurations, highlighting its robustness and effectiveness.

Table 5 compares predictive performance and memory usage across methods. While Random Fourier Features (RFF) achieves the better NLPD and MSLL, it incurs significantly higher memory cost (over 2 GB) due to dense high-dimensional features. In contrast, Nystrom methods provide more memory-efficient low-rank approximations.

Among them, the proposed Nystrom (ridge leverage) method achieves a strong accuracy–efficiency trade-off, outperforming uniform, standard leverage sampling and other baselines. Overall, these results highlight that ridge leverage Nystrom offers a more scalable alternative to RFF, delivering competitive accuracy with substantially lower memory requirements.

Table 1 highlights the significant memory advantages of the proposed Nystrom (Ridge Leverage) approach. While full-data GPR requires approximately 23.48 GB to store the kernel matrix for the Protein dataset ($n \approx 45,730$), reflecting its $\mathcal{O}(n^2)$ memory complexity, our method requires only 2.25 GB to 5.64 GB depending on the sketch size. Notably, our approach also maintains a substantially smaller memory footprint than Random Fourier Features (RFF), which consumes between 12.18 GB and 12.96 GB in this setting. This demonstrates that prioritizing statistically informative points via ridge leverage scores yields a more compact and scalable representation compared to the high-dimensional, data-independent features required by RFF to achieve comparable posterior accuracy.

Table 1: Memory usage (MB) across subset sizes on the Protein dataset (around 46K samples). All values are averaged over five random trials with standard deviations shown after $\pm$. Full-data GPR requires around 23.48GB.

| Method | 2% | 4% | 6% | 8% | 10% |
|---|---|---|---|---|---|
| **Full Data (Exact GP)** | | | 23477.29 | | |
| Uniform | $6709.6 \pm 3678.6$ | $4109.7 \pm 23.1$ | $4178.4 \pm 7.8$ | $4302.0 \pm 10.6$ | $4424.3 \pm 13.7$ |
| Leverage | $4121.3 \pm 19.1$ | $4109.2 \pm 23.5$ | $4178.3 \pm 7.6$ | $4302.0 \pm 10.6$ | $4424.1 \pm 13.6$ |
| KMeans | $4121.9 \pm 18.7$ | $4109.8 \pm 23.2$ | $4178.3 \pm 7.6$ | $4302.0 \pm 10.6$ | $4424.3 \pm 13.7$ |
| SVGP | $\mathbf{1756.8 \pm 1657.6}$ | $\mathbf{1121.5 \pm 73.2}$ | $\mathbf{1734.7 \pm 73.7}$ | $2372.0 \pm 73.3$ | $3036.2 \pm 73.9$ |
| IterGP | $2034.7 \pm 0.0$ | $2034.7 \pm 0.0$ | $2034.7 \pm 0.0$ | $\mathbf{2034.7 \pm 0.0}$ | $\mathbf{2034.7 \pm 0.0}$ |
| Nyström (Uniform) | $1953.9 \pm 75.5$ | $2842.3 \pm 79.6$ | $3753.2 \pm 84.8$ | $4687.5 \pm 89.2$ | $5644.0 \pm 93.7$ |
| Nyström (Leverage) | $2255.6 \pm 347.5$ | $2843.8 \pm 79.4$ | $3754.7 \pm 84.9$ | $4689.1 \pm 89.0$ | $5645.5 \pm 93.6$ |
| Nyström (Ridge Leverage) | $2253.5 \pm 347.7$ | $2842.4 \pm 79.8$ | $3753.7 \pm 84.9$ | $4687.7 \pm 89.2$ | $5644.3 \pm 93.7$ |
| Random Fourier Features | $12180.2 \pm 17.6$ | $12258.7 \pm 0.0$ | $12493.2 \pm 0.0$ | $12727.7 \pm 0.0$ | $12962.5 \pm 0.0$ |

Table 2: Comparison of negative log predictive density (NLPD), mean standardized log loss (MSLL) and runtime (in seconds) for Gaussian Process Regression (GPR) methods on the UCI Elevators dataset using the RBF kernel. Each experiment is repeated over 5 random seeds, and we report the mean and standard deviation across runs. Lower values are better for all metrics.

| Method | Subset % | NLPD | MSLL | Time (s) |
|---|---|---|---|---|
| | Full Data | $-0.7540$ | $-0.8224$ | $38.45$ |
| Uniform | 2% | $0.4791 \pm 0.1192$ | $0.4107 \pm 0.1192$ | $5.6802 \pm 0.0415$ |
| Leverage | 2% | $0.2869 \pm 0.0543$ | $0.2185 \pm 0.0543$ | $5.6877 \pm 0.0433$ |
| Kmeans | 2% | $0.4359 \pm 0.0824$ | $0.3675 \pm 0.0824$ | $\mathbf{5.6594 \pm 0.0470}$ |
| Svgp | 2% | $40.7380 \pm 0.2973$ | $40.6696 \pm 0.2973$ | $114.2088 \pm 0.3616$ |
| Itergp | 2% | $-0.2166 \pm 0.0000$ | $-0.2850 \pm 0.0000$ | $8.7699 \pm 0.2120$ |
| Nystrom(uniform) | 2% | $0.2323 \pm 0.0000$ | $0.1639 \pm 0.0000$ | $16.4537 \pm 0.0354$ |
| Nystrom(leverage) | 2% | $-0.1789 \pm 0.0352$ | $-0.2473 \pm 0.0352$ | $17.0823 \pm 0.0296$ |
| Nystrom(ridge leverage) | 2% | $\mathbf{-0.5551 \pm 0.0034}$ | $\mathbf{-0.6235 \pm 0.0034}$ | $21.1613 \pm 0.1409$ |
| Uniform | 4% | $0.2258 \pm 0.0428$ | $0.1574 \pm 0.0428$ | $\mathbf{6.1125 \pm 0.0073}$ |
| Leverage | 4% | $0.1558 \pm 0.0174$ | $0.0874 \pm 0.0174$ | $6.1252 \pm 0.0429$ |
| Kmeans | 4% | $0.3369 \pm 0.0516$ | $0.2685 \pm 0.0516$ | $6.1207 \pm 0.0113$ |
| Svgp | 4% | $41.6886 \pm 0.2967$ | $41.6202 \pm 0.2967$ | $116.2481 \pm 0.1822$ |
| Itergp | 4% | $-0.3124 \pm 0.0000$ | $-0.3808 \pm 0.0000$ | $22.2070 \pm 0.1170$ |
| Nystrom(uniform) | 4% | $-0.4713 \pm 0.0492$ | $-0.5397 \pm 0.0492$ | $18.6045 \pm 0.0160$ |
| Nystrom(leverage) | 4% | $-0.3172 \pm 0.0174$ | $-0.3856 \pm 0.0174$ | $19.0833 \pm 0.0375$ |
| Nystrom(ridge leverage) | 4% | $\mathbf{-0.6380 \pm 0.0032}$ | $\mathbf{-0.7064 \pm 0.0032}$ | $21.7761 \pm 0.0655$ |
| Uniform | 6% | $0.1517 \pm 0.0509$ | $0.0833 \pm 0.0509$ | $6.4596 \pm 0.0220$ |
| Leverage | 6% | $0.1157 \pm 0.0173$ | $0.0473 \pm 0.0173$ | $6.4745 \pm 0.0210$ |
| Kmeans | 6% | $0.3647 \pm 0.1634$ | $0.2963 \pm 0.1634$ | $\mathbf{6.4564 \pm 0.0371}$ |
| Svgp | 6% | $40.0423 \pm 0.1234$ | $39.9739 \pm 0.1234$ | $118.1078 \pm 0.2941$ |
| Itergp | 6% | $-0.3415 \pm 0.0000$ | $-0.4099 \pm 0.0000$ | $43.3009 \pm 0.0840$ |
| Nystrom(uniform) | 6% | $-0.5925 \pm 0.0074$ | $-0.6609 \pm 0.0074$ | $20.4320 \pm 0.0431$ |
| Nystrom(leverage) | 6% | $-0.4420 \pm 0.0122$ | $-0.5104 \pm 0.0122$ | $20.9021 \pm 0.0344$ |
| Nystrom(ridge leverage) | 6% | $\mathbf{-0.6815 \pm 0.0021}$ | $\mathbf{-0.7499 \pm 0.0021}$ | $22.7822 \pm 0.1542$ |
| Uniform | 8% | $0.1517 \pm 0.0308$ | $0.0833 \pm 0.0308$ | $\mathbf{7.1314 \pm 0.0217}$ |
| Leverage | 8% | $0.0987 \pm 0.0140$ | $0.0303 \pm 0.0140$ | $7.2971 \pm 0.0233$ |
| Kmeans | 8% | $0.3060 \pm 0.0824$ | $0.2376 \pm 0.0824$ | $7.2365 \pm 0.0169$ |
| Svgp | 8% | $38.7838 \pm 0.2665$ | $38.7154 \pm 0.2665$ | $122.1149 \pm 0.1595$ |
| Itergp | 8% | $-0.3629 \pm 0.0000$ | $-0.4313 \pm 0.0000$ | $70.7370 \pm 0.2089$ |
| Nystrom(uniform) | 8% | $-0.6465 \pm 0.0031$ | $-0.7149 \pm 0.0031$ | $20.5632 \pm 0.0910$ |
| Nystrom(leverage) | 8% | $-0.5343 \pm 0.0088$ | $-0.6027 \pm 0.0088$ | $21.2114 \pm 0.2373$ |
| Nystrom(ridge leverage) | 8% | $\mathbf{-0.7115 \pm 0.0021}$ | $\mathbf{-0.7799 \pm 0.0021}$ | $22.5875 \pm 0.1829$ |
| Uniform | 10% | $0.1202 \pm 0.0399$ | $0.0518 \pm 0.0399$ | $\mathbf{7.1129 \pm 0.0575}$ |
| Leverage | 10% | $0.0541 \pm 0.0274$ | $-0.0143 \pm 0.0274$ | $7.2908 \pm 0.0144$ |
| Kmeans | 10% | $0.2613 \pm 0.0984$ | $0.1929 \pm 0.0984$ | $7.2396 \pm 0.0129$ |
| Svgp | 10% | $37.5595 \pm 0.0773$ | $37.4911 \pm 0.0773$ | $127.6543 \pm 0.3152$ |
| Itergp | 10% | $-0.3798 \pm 0.0000$ | $-0.4482 \pm 0.0000$ | $104.1623 \pm 0.1610$ |
| Nystrom(uniform) | 10% | $-0.6804 \pm 0.0022$ | $-0.7488 \pm 0.0022$ | $22.1774 \pm 0.0991$ |
| Nystrom(leverage) | 10% | $-0.5982 \pm 0.0048$ | $-0.6666 \pm 0.0048$ | $22.7742 \pm 0.0430$ |
| Nystrom(ridge leverage) | 10% | $\mathbf{-0.7323 \pm 0.0024}$ | $\mathbf{-0.8007 \pm 0.0024}$ | $23.9634 \pm 0.0825$ |

Table 3: Comparison of negative log predictive density (NLPD), mean standardized log loss (MSLL) and memory (in MB) for Gaussian Process Regression (GPR) methods on the UCI California Housing dataset using the Matern kernel. Each experiment is repeated over 5 random seeds, and we report the mean and standard deviation across runs. Lower values are better for all metrics.

| Method | Subset % | NLPD | MSLL | Memory (MB) |
|---|---|---|---|---|
| | Full Data | 0.8987 | $-0.6563$ | 4797.03 |
| Uniform | 2% | $2.0318 \pm 0.1416$ | $0.4768 \pm 0.1416$ | $1169.2865 \pm 638.5115$ |
| Leverage | 2% | $1.8690 \pm 0.0424$ | $0.3140 \pm 0.0424$ | $853.2596 \pm 6.4547$ |
| Kmeans | 2% | $2.0137 \pm 0.1869$ | $0.4587 \pm 0.1869$ | $854.2215 \pm 6.4547$ |
| Svgp | 2% | $10.9006 \pm 0.1611$ | $9.3456 \pm 0.1611$ | $\mathbf{278.0277 \pm 291.5613}$ |
| Itergp | 2% | $1.0109 \pm 0.0000$ | $-0.5442 \pm 0.0000$ | $304.2607 \pm 0.0000$ |
| Nystrom(uniform) | 2% | $1.1239 \pm 0.0189$ | $-0.4311 \pm 0.0189$ | $507.2209 \pm 13.1754$ |
| Rff | 2% | $\mathbf{0.8915 \pm 0.0068}$ | $\mathbf{-0.6635 \pm 0.0068}$ | $2473.1115 \pm 0.0191$ |
| Nystrom(ridge leverage) | 2% | $1.0696 \pm 0.0072$ | $-0.4854 \pm 0.0072$ | $552.9800 \pm 77.3543$ |
| Uniform | 4% | $2.0952 \pm 0.1466$ | $0.5402 \pm 0.1466$ | $887.7567 \pm 6.9622$ |
| Leverage | 4% | $1.9214 \pm 0.0978$ | $0.3663 \pm 0.0978$ | $887.6729 \pm 6.9197$ |
| Kmeans | 4% | $2.0195 \pm 0.1899$ | $0.4645 \pm 0.1899$ | $887.7592 \pm 6.9622$ |
| Svgp | 4% | $10.7327 \pm 0.1574$ | $9.1777 \pm 0.1574$ | $\mathbf{245.7788 \pm 12.9210}$ |
| Itergp | 4% | $0.9691 \pm 0.0000$ | $-0.5859 \pm 0.0000$ | $304.2607 \pm 0.0000$ |
| Nystrom(uniform) | 4% | $0.9720 \pm 0.0076$ | $-0.5830 \pm 0.0076$ | $723.8914 \pm 13.8915$ |
| Rff | 4% | $\mathbf{0.8688 \pm 0.0034}$ | $\mathbf{-0.6862 \pm 0.0034}$ | $2520.9160 \pm 0.0000$ |
| Nystrom(ridge leverage) | 4% | $0.9479 \pm 0.0075$ | $-0.6071 \pm 0.0075$ | $724.5246 \pm 13.9494$ |
| Uniform | 6% | $2.0469 \pm 0.0724$ | $0.4918 \pm 0.0724$ | $873.5581 \pm 8.3471$ |
| Leverage | 6% | $1.8013 \pm 0.0543$ | $0.2462 \pm 0.0543$ | $873.7120 \pm 8.4859$ |
| Kmeans | 6% | $1.9992 \pm 0.1683$ | $0.4441 \pm 0.1683$ | $874.0355 \pm 8.1051$ |
| Svgp | 6% | $10.3749 \pm 0.1324$ | $8.8199 \pm 0.1324$ | $371.1588 \pm 12.8385$ |
| Itergp | 6% | $0.9521 \pm 0.0000$ | $-0.6029 \pm 0.0000$ | $\mathbf{304.2607 \pm 0.0000}$ |
| Nystrom(uniform) | 6% | $0.9040 \pm 0.0074$ | $-0.6510 \pm 0.0074$ | $948.4028 \pm 14.9609$ |
| Rff | 6% | $\mathbf{0.8546 \pm 0.0019}$ | $\mathbf{-0.7005 \pm 0.0019}$ | $2568.7104 \pm 0.0000$ |
| Nystrom(ridge leverage) | 6% | $0.8791 \pm 0.0059$ | $-0.6759 \pm 0.0059$ | $948.4933 \pm 15.0164$ |
| Uniform | 8% | $1.9365 \pm 0.0553$ | $0.3815 \pm 0.0553$ | $889.3438 \pm 1.6545$ |
| Leverage | 8% | $1.7552 \pm 0.0312$ | $0.2002 \pm 0.0312$ | $889.0636 \pm 1.9553$ |
| Kmeans | 8% | $1.9944 \pm 0.1438$ | $0.4394 \pm 0.1438$ | $888.8565 \pm 1.6863$ |
| Svgp | 8% | $10.0673 \pm 0.1195$ | $8.5122 \pm 0.1195$ | $499.9782 \pm 12.4930$ |
| Itergp | 8% | $0.9474 \pm 0.0000$ | $-0.6076 \pm 0.0000$ | $\mathbf{304.2607 \pm 0.0000}$ |
| Nystrom(uniform) | 8% | $0.8518 \pm 0.0039$ | $-0.7032 \pm 0.0039$ | $1177.9729 \pm 15.1928$ |
| Rff | 8% | $0.8548 \pm 0.0026$ | $-0.7002 \pm 0.0026$ | $2616.5054 \pm 0.0000$ |
| Nystrom(ridge leverage) | 8% | $\mathbf{0.8387 \pm 0.0017}$ | $\mathbf{-0.7164 \pm 0.0017}$ | $1177.5941 \pm 15.3684$ |
| Uniform | 12% | $1.8548 \pm 0.0359$ | $0.2998 \pm 0.0359$ | $934.2386 \pm 5.1368$ |
| Leverage | 12% | $1.6833 \pm 0.0524$ | $0.1283 \pm 0.0524$ | $934.4450 \pm 5.2379$ |
| Kmeans | 12% | $1.8868 \pm 0.0508$ | $0.3318 \pm 0.0508$ | $934.0741 \pm 5.3000$ |
| Svgp | 12% | $9.6147 \pm 0.1137$ | $8.0597 \pm 0.1137$ | $769.6117 \pm 25.6772$ |
| Itergp | 12% | $0.9388 \pm 0.0000$ | $-0.6162 \pm 0.0000$ | $\mathbf{304.2607 \pm 0.0000}$ |
| Nystrom(uniform) | 12% | $0.8121 \pm 0.0012$ | $-0.7429 \pm 0.0012$ | $1648.8987 \pm 33.2553$ |
| Rff | 12% | $0.8455 \pm 0.0042$ | $-0.7095 \pm 0.0042$ | $2712.0942 \pm 0.0000$ |
| Nystrom(ridge leverage) | 12% | $\mathbf{0.8103 \pm 0.0009}$ | $\mathbf{-0.7447 \pm 0.0009}$ | $1649.0237 \pm 33.2553$ |

Table 4: Comparison of negative log predictive density (NLPD), mean standardized log loss (MSLL) and memory usage (in megabytes) for Gaussian Process Regression (GPR) methods on the UCI Airfoil Self-Noise dataset using the Matern kernel. Results are averaged over 5 random seeds, with standard deviations reported.

| Method | Subset % | NLPD | MSLL | Memory (MB) |
|---|---|---|---|---|
| | Full Data | 2.4025 | $-0.9750$ | 51.12 |
| Uniform | 2% | $4.1267 \pm 0.3406$ | $0.7492 \pm 0.3406$ | $26.6922 \pm 5.0963$ |
| Leverage | 2% | $3.8720 \pm 0.0404$ | $0.4945 \pm 0.0404$ | $26.1291 \pm 0.1068$ |
| Kmeans | 2% | $3.9280 \pm 0.1716$ | $0.5505 \pm 0.1716$ | $24.1977 \pm 0.1063$ |
| Svgp | 2% | $4.7646 \pm 0.0660$ | $1.3871 \pm 0.0660$ | $19.9855 \pm 2.2387$ |
| Itergp | 2% | $3.0549 \pm 0.0000$ | $-0.3225 \pm 0.0000$ | $\mathbf{19.2832 \pm 0.0000}$ |
| Nystrom(uniform) | 2% | $2.8919 \pm 0.3355$ | $-0.4856 \pm 0.3355$ | $20.7281 \pm 0.1014$ |
| Nystrom(leverage) | 2% | $3.5628 \pm 0.0000$ | $0.1853 \pm 0.0000$ | $22.7260 \pm 0.4168$ |
| Rff | 2% | $3.4720 \pm 0.1643$ | $0.0945 \pm 0.1643$ | $53.0475 \pm 0.0027$ |
| Nystrom(ridge leverage) | 2% | $\mathbf{2.6845 \pm 0.0065}$ | $\mathbf{-0.6929 \pm 0.0065}$ | $20.9984 \pm 0.4168$ |
| Uniform | 4% | $3.9302 \pm 0.1720$ | $0.5527 \pm 0.1720$ | $24.2454 \pm 0.0131$ |
| Leverage | 4% | $3.7505 \pm 0.0482$ | $0.3730 \pm 0.0482$ | $26.1022 \pm 0.0133$ |
| Kmeans | 4% | $3.8046 \pm 0.0635$ | $0.4272 \pm 0.0635$ | $24.2459 \pm 0.0131$ |
| Svgp | 4% | $4.6540 \pm 0.0517$ | $1.2765 \pm 0.0517$ | $20.4402 \pm 0.1234$ |
| Itergp | 4% | $2.8807 \pm 0.0000$ | $-0.4968 \pm 0.0000$ | $\mathbf{19.2832 \pm 0.0000}$ |
| Nystrom(uniform) | 4% | $2.6060 \pm 0.0017$ | $-0.7714 \pm 0.0017$ | $22.2289 \pm 0.0959$ |
| Nystrom(leverage) | 4% | $2.6762 \pm 0.0110$ | $-0.7013 \pm 0.0110$ | $22.7094 \pm 0.0959$ |
| Rff | 4% | $2.8534 \pm 0.0556$ | $-0.5241 \pm 0.0556$ | $53.0543 \pm 0.0027$ |
| Nystrom(ridge leverage) | 4% | $\mathbf{2.6024 \pm 0.0063}$ | $\mathbf{-0.7751 \pm 0.0063}$ | $22.2401 \pm 0.0959$ |
| Uniform | 6% | $3.7681 \pm 0.0825$ | $0.3906 \pm 0.0825$ | $24.3877 \pm 0.0225$ |
| Leverage | 6% | $3.6811 \pm 0.0328$ | $0.3037 \pm 0.0328$ | $26.1542 \pm 0.0227$ |
| Kmeans | 6% | $3.8331 \pm 0.1226$ | $0.4556 \pm 0.1226$ | $24.3882 \pm 0.0225$ |
| Svgp | 6% | $4.3329 \pm 0.0323$ | $0.9554 \pm 0.0323$ | $22.1729 \pm 0.1336$ |
| Itergp | 6% | $2.8622 \pm 0.0000$ | $-0.5153 \pm 0.0000$ | $\mathbf{19.2832 \pm 0.0000}$ |
| Nystrom(uniform) | 6% | $2.5551 \pm 0.0058$ | $-0.8224 \pm 0.0058$ | $23.7790 \pm 0.1021$ |
| Nystrom(leverage) | 6% | $2.5932 \pm 0.0074$ | $-0.7842 \pm 0.0074$ | $23.8132 \pm 0.1021$ |
| Rff | 6% | $2.7679 \pm 0.0901$ | $-0.6096 \pm 0.0901$ | $53.0650 \pm 0.0047$ |
| Nystrom(ridge leverage) | 6% | $\mathbf{2.5498 \pm 0.0046}$ | $\mathbf{-0.8277 \pm 0.0046}$ | $23.7907 \pm 0.1021$ |
| Uniform | 8% | $3.6592 \pm 0.0534$ | $0.2817 \pm 0.0534$ | $24.5618 \pm 0.0307$ |
| Leverage | 8% | $3.6385 \pm 0.0174$ | $0.2610 \pm 0.0174$ | $26.2272 \pm 0.0309$ |
| Kmeans | 8% | $3.7262 \pm 0.1350$ | $0.3488 \pm 0.1350$ | $24.5623 \pm 0.0307$ |
| Svgp | 8% | $4.2478 \pm 0.0368$ | $0.8703 \pm 0.0368$ | $23.9900 \pm 0.1410$ |
| Itergp | 8% | $2.8677 \pm 0.0000$ | $-0.5098 \pm 0.0000$ | $\mathbf{19.2832 \pm 0.0000}$ |
| Nystrom(uniform) | 8% | $2.5166 \pm 0.0032$ | $-0.8609 \pm 0.0032$ | $25.3638 \pm 0.1064$ |
| Nystrom(leverage) | 8% | $2.5431 \pm 0.0058$ | $-0.8344 \pm 0.0058$ | $25.3979 \pm 0.1064$ |
| Rff | 8% | $2.7147 \pm 0.0553$ | $-0.6628 \pm 0.0553$ | $53.0799 \pm 0.0062$ |
| Nystrom(ridge leverage) | 8% | $\mathbf{2.5154 \pm 0.0039}$ | $\mathbf{-0.8621 \pm 0.0039}$ | $25.3755 \pm 0.1064$ |
| Uniform | 10% | $3.5781 \pm 0.0542$ | $0.2007 \pm 0.0542$ | $24.7744 \pm 0.0400$ |
| Leverage | 10% | $3.5993 \pm 0.0197$ | $0.2218 \pm 0.0197$ | $26.3231 \pm 0.0402$ |
| Kmeans | 10% | $3.6498 \pm 0.1140$ | $0.2723 \pm 0.1140$ | $24.7749 \pm 0.0400$ |
| Svgp | 10% | $4.0887 \pm 0.0631$ | $0.7112 \pm 0.0631$ | $25.9065 \pm 0.1508$ |
| Itergp | 10% | $2.8104 \pm 0.0000$ | $-0.5671 \pm 0.0000$ | $\mathbf{19.2832 \pm 0.0000}$ |
| Nystrom(uniform) | 10% | $2.4918 \pm 0.0028$ | $-0.8857 \pm 0.0028$ | $26.9956 \pm 0.1123$ |
| Nystrom(leverage) | 10% | $2.5148 \pm 0.0021$ | $-0.8627 \pm 0.0021$ | $27.0298 \pm 0.1123$ |
| Rff | 10% | $2.6256 \pm 0.0686$ | $-0.7519 \pm 0.0686$ | $53.0994 \pm 0.0082$ |
| Nystrom(ridge leverage) | 10% | $\mathbf{2.4890 \pm 0.0124}$ | $\mathbf{-0.8915 \pm 0.0154}$ | $27.0073 \pm 0.1123$ |

Table 5: Comparison of negative log predictive density (NLPD), mean standardized log loss (MSLL) and memory usage (in megabytes) for Gaussian Process Regression (GPR) methods on the UCI Elevators dataset using the Matern kernel. Results are averaged over 5 random seeds, with standard deviations reported.

| Method | Subset % | NLPD | MSLL | Memory (MB) |
|---|---|---|---|---|
| | Full Data | $-0.8485$ | $-0.9169$ | **4055.47** |
| Uniform | 3% | $0.3001 \pm 0.0167$ | $0.2317 \pm 0.0167$ | $247.6358 \pm 67.6086$ |
| Leverage | 3% | $0.2597 \pm 0.0188$ | $0.1913 \pm 0.0188$ | $215.1867 \pm 2.7777$ |
| Kmeans | 3% | $0.4627 \pm 0.1138$ | $0.3943 \pm 0.1138$ | $215.1867 \pm 2.7777$ |
| Svgp | 3% | $4.5856 \pm 0.0548$ | $4.5172 \pm 0.0548$ | **161.6910 ± 16.6343** |
| Itergp | 3% | $-0.1158 \pm 0.0000$ | $-0.1842 \pm 0.0000$ | $224.2437 \pm 0.0000$ |
| Nystrom(uniform) | 3% | $-0.7421 \pm 0.0086$ | $-0.8105 \pm 0.0086$ | $464.0999 \pm 16.9594$ |
| Rff | 3% | $\mathbf{-0.9004 \pm 0.0023}$ | $\mathbf{-0.9688 \pm 0.0023}$ | $2114.6526 \pm 0.0102$ |
| Nystrom(ridge leverage) | 3% | $-0.7195 \pm 0.0092$ | $-0.7879 \pm 0.0092$ | $482.6678 \pm 16.9594$ |
| Uniform | 5% | $0.2454 \pm 0.0407$ | $0.1770 \pm 0.0407$ | $236.0326 \pm 4.4180$ |
| Leverage | 5% | $0.2018 \pm 0.0127$ | $0.1334 \pm 0.0127$ | $236.0498 \pm 4.4249$ |
| Kmeans | 5% | $0.3895 \pm 0.0902$ | $0.3211 \pm 0.0902$ | $236.0340 \pm 4.4216$ |
| Svgp | 5% | $4.5543 \pm 0.0655$ | $4.4859 \pm 0.0655$ | $272.8118 \pm 11.1136$ |
| Itergp | 5% | $-0.1522 \pm 0.0000$ | $-0.2206 \pm 0.0000$ | **224.2437 ± 0.0000** |
| Nystrom(uniform) | 5% | $-0.7845 \pm 0.0042$ | $-0.8529 \pm 0.0042$ | $655.1745 \pm 12.2674$ |
| Rff | 5% | $\mathbf{-0.9048 \pm 0.0012}$ | $\mathbf{-0.9732 \pm 0.0012}$ | $2154.9487 \pm 0.0000$ |
| Nystrom(ridge leverage) | 5% | $-0.7510 \pm 0.0042$ | $-0.8195 \pm 0.0042$ | $674.4363 \pm 11.9642$ |
| Uniform | 6% | $0.2182 \pm 0.0321$ | $0.1498 \pm 0.0321$ | $250.7646 \pm 2.6318$ |
| Leverage | 6% | $0.1745 \pm 0.0132$ | $0.1061 \pm 0.0132$ | $251.1238 \pm 2.7003$ |
| Kmeans | 6% | $0.3857 \pm 0.1038$ | $0.3173 \pm 0.1038$ | $250.7249 \pm 2.6072$ |
| Svgp | 6% | $4.5257 \pm 0.0321$ | $4.4573 \pm 0.0321$ | $327.4325 \pm 5.6037$ |
| Itergp | 6% | $-0.1639 \pm 0.0000$ | $-0.2323 \pm 0.0000$ | **224.2437 ± 0.0000** |
| Nystrom(uniform) | 6% | $-0.7980 \pm 0.0049$ | $-0.8664 \pm 0.0049$ | $750.4968 \pm 6.0018$ |
| Rff | 6% | $\mathbf{-0.9062 \pm 0.0024}$ | $\mathbf{-0.9746 \pm 0.0024}$ | $2175.1685 \pm 0.0000$ |
| Nystrom(ridge leverage) | 6% | $-0.7648 \pm 0.0015$ | $-0.8332 \pm 0.0015$ | $770.0260 \pm 6.0896$ |
| Uniform | 9% | $0.1695 \pm 0.0139$ | $0.1011 \pm 0.0139$ | $238.6730 \pm 5.9560$ |
| Leverage | 9% | $0.1019 \pm 0.0286$ | $0.0335 \pm 0.0286$ | $238.8654 \pm 6.2307$ |
| Kmeans | 9% | $0.2867 \pm 0.0771$ | $0.2183 \pm 0.0771$ | $238.5708 \pm 5.9930$ |
| Svgp | 9% | $4.3862 \pm 0.0311$ | $4.3178 \pm 0.0311$ | $494.8212 \pm 15.9441$ |
| Itergp | 9% | $-0.1921 \pm 0.0000$ | $-0.2605 \pm 0.0000$ | **224.2437 ± 0.0000** |
| Nystrom(uniform) | 9% | $-0.8212 \pm 0.0039$ | $-0.8896 \pm 0.0039$ | $1036.9442 \pm 19.7361$ |
| Rff | 9% | $\mathbf{-0.9045 \pm 0.0037}$ | $\mathbf{-0.9729 \pm 0.0037}$ | $2235.8320 \pm 0.0000$ |
| Nystrom(ridge leverage) | 9% | $-0.7907 \pm 0.0008$ | $-0.8591 \pm 0.0008$ | $1055.7627 \pm 19.2402$ |
| Uniform | 10% | $0.1382 \pm 0.0212$ | $0.0698 \pm 0.0212$ | $242.4258 \pm 0.8025$ |
| Leverage | 10% | $0.0663 \pm 0.0291$ | $-0.0021 \pm 0.0291$ | $242.4521 \pm 0.6862$ |
| Kmeans | 10% | $0.2509 \pm 0.0840$ | $0.1825 \pm 0.0840$ | $242.2792 \pm 1.2604$ |
| Svgp | 10% | $4.3341 \pm 0.0123$ | $4.2657 \pm 0.0123$ | $556.8342 \pm 4.8722$ |
| Itergp | 10% | $-0.1979 \pm 0.0000$ | $-0.2663 \pm 0.0000$ | **224.2437 ± 0.0000** |
| Nystrom(uniform) | 10% | $-0.8249 \pm 0.0035$ | $-0.8933 \pm 0.0035$ | $1151.1535 \pm 7.5045$ |
| Rff | 10% | $\mathbf{-0.9079 \pm 0.0018}$ | $\mathbf{-0.9763 \pm 0.0018}$ | $2256.6812 \pm 0.0000$ |
| Nystrom(ridge leverage) | 10% | $-0.7977 \pm 0.0025$ | $-0.8661 \pm 0.0025$ | $1169.7740 \pm 7.4129$ |

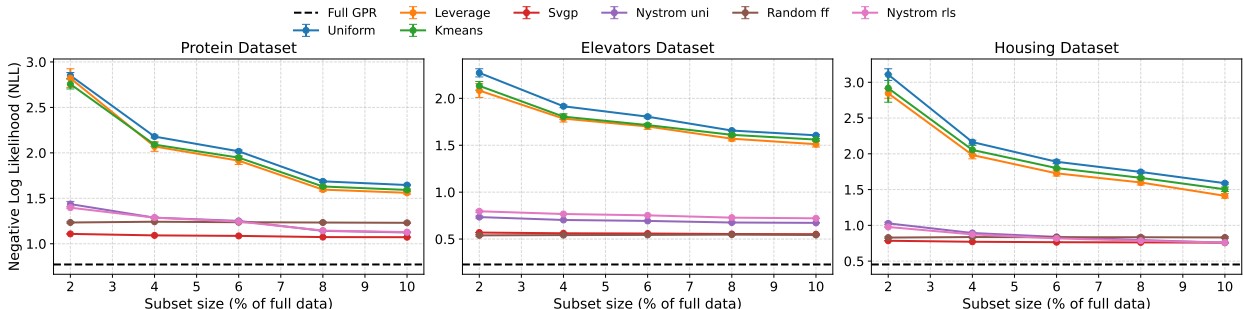

Figure 5: Comparison of Negative Log Likelihood (NLL) across different subset sizes and Gaussian Process approximation methods on the Protein, Elevators, and Housing datasets using the Matern kernel. Each subplot reports the mean and standard deviation over five random trials. The dashed horizontal line denotes the performance of the full-data (Exact GP) model.

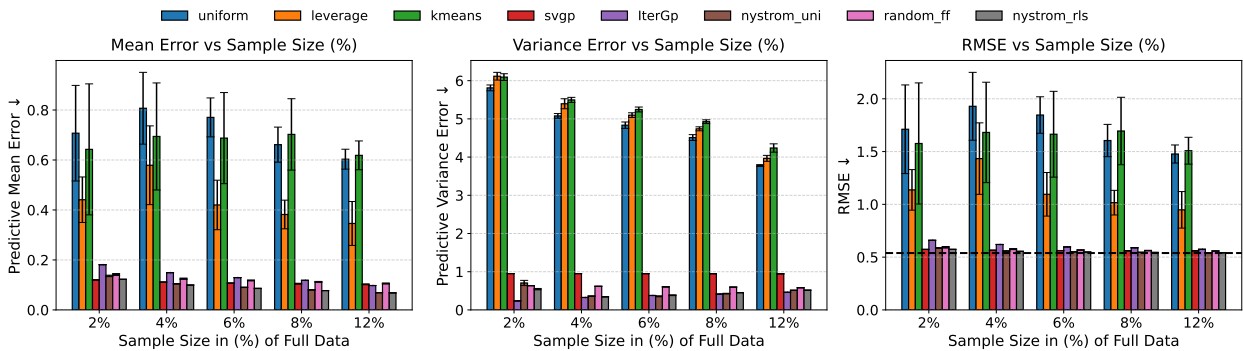

Figure 6: **Results on California Housing Dataset.** Evaluation of Gaussian Process Regression methods on the Housing dataset using the Matern kernel. Predictive mean error, predictive variance error, and RMSE are plotted versus subset size. Ridge leverage–based GPR yields the best tradeoff across metrics. All results are averaged over 5 random trials, with standard deviations shown as error bars. The dashed horizontal line indicates the performance of the full-dataset (exact GP) model.

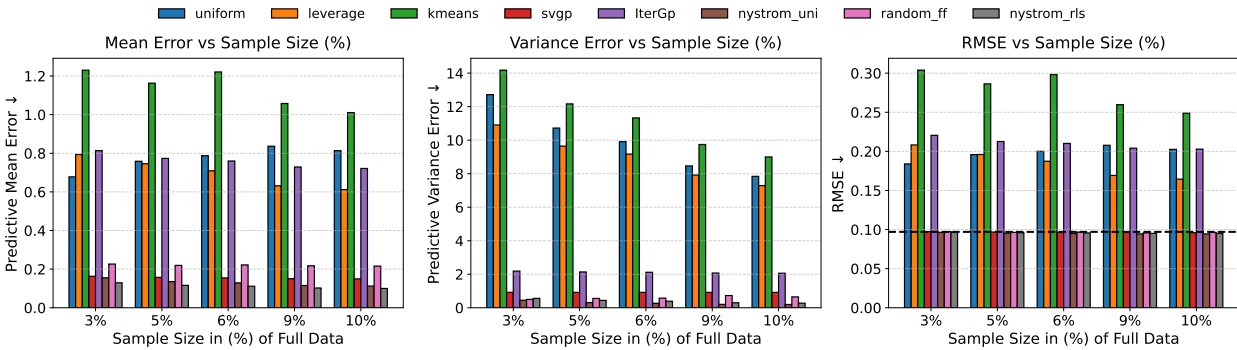

Figure 7: **Results on UCI Elevators Dataset.** Evaluation of Gaussian Process Regression methods on the UCI Elevators dataset using the Matern kernel. Predictive mean error, predictive variance error, and RMSE are plotted versus subset size. Ridge leverage–based GPR yields the best tradeoff across metrics. All results are averaged over 5 random trials, with standard deviations shown as error bars. The dashed horizontal line indicates the performance of the full-dataset (exact GP) model.

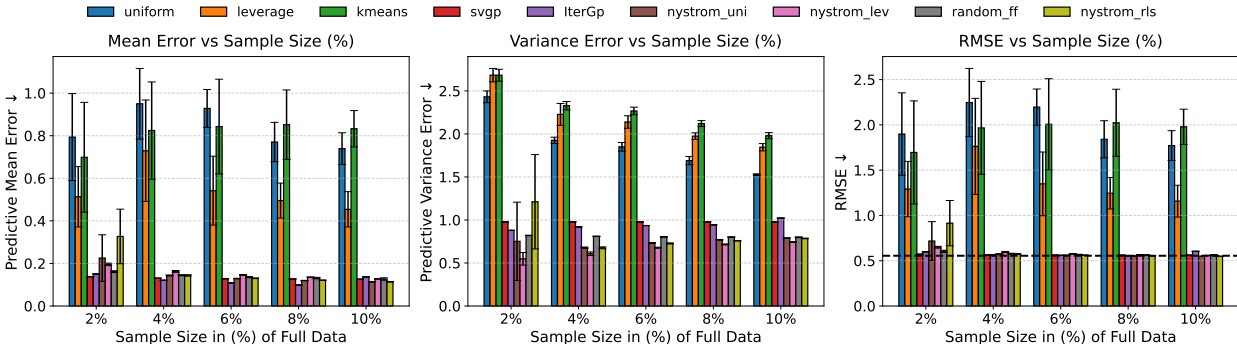

Figure 8: **Results on California Housing Dataset.** Evaluation of Gaussian Process Regression methods on the Housing dataset using the **RBF kernel**. Predictive mean error, predictive variance error, and RMSE are plotted versus subset size. Ridge Leverage based GPR yields the best tradeoff across metrics. All results are averaged over 5 random trials with standard deviation shown as error bars. The dashed horizontal line indicates the performance of the full-dataset (exact GP) model.

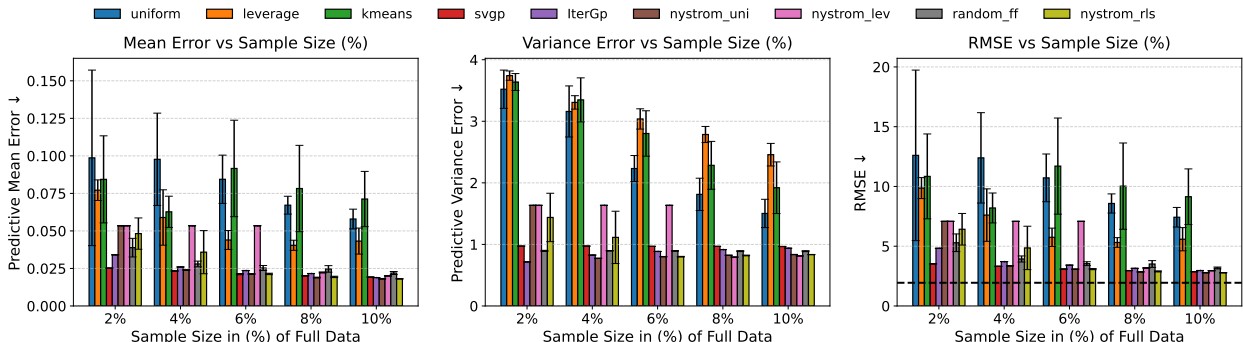

Figure 9: **Results on Airfoil Self-Noise Dataset.** Performance of various GPR methods on the Airfoil dataset using the RBF kernel. Ridge leverage−based sketching again outperforms uniform, SVGP and RFF in both predictive accuracy and variance estimation. All results are averaged over 5 random trials with standard deviation shown as error bars. The dashed horizontal line indicates the performance of the full-dataset (exact GP) model.

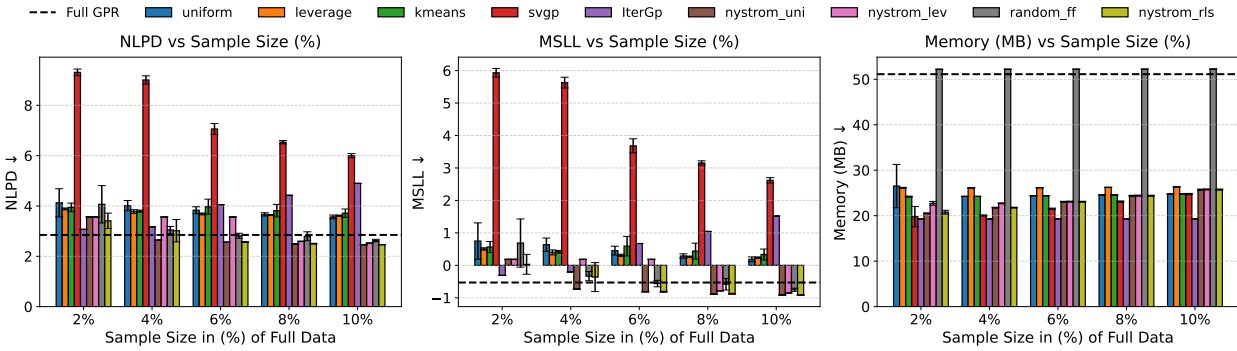

Figure 10: **Results on Airfoil Self-Noise Dataset.** Performance of various GPR methods on the *Airfoil* dataset using the RBF kernel. All results are averaged over 5 random trials, with standard deviations shown as error bars. The dashed horizontal line indicates the performance of the full-dataset (exact GP) model. Due to the relatively small dataset size, the memory usage of the proposed and baseline approximation methods remains nearly constant across subset sizes from 2% to 10%. Nevertheless, all approximation methods consume substantially less memory than the full GP, highlighting their efficiency. RFF requires substantially higher memory to match the posterior accuracy achieved by our method.

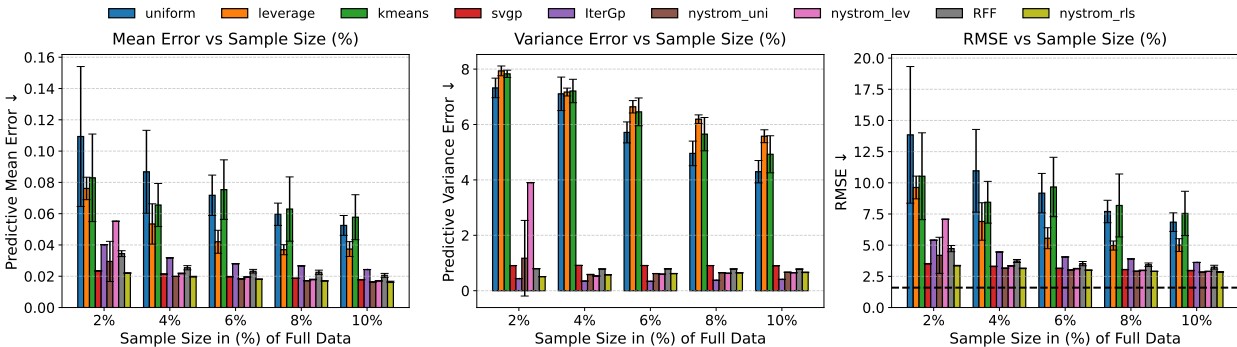

Figure 11: **Results on Airfoil Self-Noise Dataset.** Performance of various Gaussian Process Regression (GPR) methods on the UCI Airfoil Self-Noise dataset using the Matern kernel. Ridge leverage–based sketching consistently outperforms uniform sampling and SVGP in both predictive accuracy and variance estimation. All results are averaged over 5 random trials, with standard deviations shown as error bars. The dashed horizontal line indicates the performance of the full-dataset (exact GP) model.

