# OpenReview forum: "On Sketching for Gaussian Process Regression with New Statistical Guarantees"
_TMLR — Accepted by TMLR_

### Review · Reviewer_vqEv · 2025-11-18

**Summary Of Contributions:**

The paper tackle the scalability of Gaussian Processes to large datasets using sketching. Specifically, the sketching scheme proposed relies on ridge leverage score sub-sampling, following El Alaoui & Mahoney (2015). It provides theoretical guarantees together with empirical experiments. The main contributions of the paper are:
1) theoretically, the authors provide learning bounds for the sketched posterior mean, posterior variance and Negative Log Marginal Likelihood;
2) empirically, the authors provide experiments on real-world datasets to demonstrate the relevance of their method.

Strengths:
- The paper is rather well written, except the theoretical guarantees where the assumptions are not clearly stated and some discussions are missing.
- The paper is clearly motivated.
- The claims are supported by theoretical and empirical evidences.

Weaknesses:
- The contributions of this paper are incremental results following El Alaoui & Mahoney (2015), but this should not be an issue for TMLR.
- The related works should mention Random Fourier Features.
- The theoretical guarantees lack discussions.
- The paper lacks some discussions on the limitations of the frameworks and future work directions.

**Audience:**

Yes

**Audience Explanation:**

I think it would interest the TMLR's audience as GPs are a very famous Bayesian framework for Machine Learning tasks and their expensive cost with respect to the number of training data is a major bottleneck.

**Claims And Evidence:**

Yes

**Claims Explanation:**

The theoretical claims are supported with sound proofs. However, I have a few remarks regarding the theory section and requested changes (see Requested Changes section).

Empirically, the authors provide a comparison with many baselines on real-worlds GP Regression datasets. However, I think that it would be relevant to add Random Fourier Features as a baseline, since it is the most standard kernel approximation technique together with sketching. Moreover, I have a question regarding the "sample size" in the figures' x-axis: Is it, in your case, the sketching size m? Or does that mean that you use small percentage of the full training dataset as training points? It should be clarified.

**Requested Changes:**

A) Related works:

1) Add a small discussion with Random Fourier Features since it is the other famous method to scale kernel methods by approximating the kernel itself directly, similarly to Nystrom approximation or sketching. In which cases would you rather use sketching over RFF (in terms of data or kernel at hand)?

B) Theoretical guarantees:

1) Assumptions: it seems that you consider two assumptions (on (i) $K-L_{\gamma}$ and (ii) $k_{\*}$), you should declare them clearly and discuss them. For assumption (ii), does it mean that for any test point $x^{\*}$, there exists such a vector $\alpha$? Then maybe you should use $\alpha^{\*}$ as notation.

2) You should move the remark (Interpretation of α in the Predictive Mean and Variance Bounds) earlier to discuss assumption (ii). To what extent such an assumption is realistic? Do you have any specific examples?

3) Your theorems lack discussions and interpretations. In particular, theorems 2 and 4 are not discussed at all. How theorem 2 compare with existing results?

4) Role of \beta? It seems like we would like it to be close to 1 but then the $p_{i}$s seem unrealistic. Could you elaborate a bit more?

5) Could you provide learning rates with respect to $n$? You could derive a Corollary assuming a polynomial eigendecay for instance and specific settings for every parameters and hyperparameters ($\alpha$, $\gamma$, $\sigma_{\xi}$, $t$...) to obtain a final rate.

C) Experiments

1) Since you derive learning bounds, it would be nice to provide synthetic experiments in which you control the assumptions considered, such as the eigendecay of the kernel or this $\alpha$ to show that there exist scenarios in which these assumptions make sense, and retrieve the theoretical results on an empirical example.

2) "sample size" in the figures' x-axis: Is it, in your case, the sketching size m? Or does that mean that you use small percentage of the full training dataset as training points? It should be clarified.

3) You do not provide figures showcasing the training time of the different methods. For instance, a figure with the evaluation metrics w.r.t. training time would be relevant.

---

> ### Author Response · Authors · 2026-01-09
> **Data-Adaptive Sketching vs. RFF: Theory and Empirical Evidence for Superior Uncertainty Quantification**
>
> We thank the reviewer for their constructive feedback and positive assessment. We address all requested changes below and will incorporate the suggested improvements in the revised manuscript.
>
> ---
>
> ### A) Related Work: Random Fourier Features (RFF)
>
> We added an empirical comparison with Random Fourier Features (RFF). RFF approximates kernels via data-independent random features, whereas our method uses data-dependent ridge leverage score Nystrom sketching.
> (i) RFF requires storing dense $n\times D$ feature matrices, leading to higher memory usage for comparable accuracy, while our method stores only $n\times m$ kernel factors.
> (ii) Ridge leverage sketching achieves substantially lower predictive variance error, indicating improved preservation of posterior uncertainty.
> (iii) RFF ignores structure of data, whereas our method exploits non-uniformity in the dataset.
>
> Overall, RFF is preferable when kernel evaluation is expensive, while ridge leverage sketching is more suitable when memory is constrained or when accurate uncertainty calibration is critical.
>
> ---
>
> ### B) Theoretical Guarantees
>
> **B.1 Assumptions on $K-L_\gamma$ and $k_*$.**
> We will explicitly restate these assumptions earlier in the theory section. The bound $0\preceq K-L_\gamma\preceq \frac{\gamma}{1-t}I$ follows directly from ridge leverage score Nystrom analysis (El Alaoui and Mahoney, 2015). The representation $k_*=U\alpha$ is purely a change of basis and always exists with $\alpha=U^\top k_*$. We will change notation to $\alpha^*$.
>
> **B.2 Interpretation and realism of $\alpha$.**
> We agree with the reviewer’s suggestion and will move the discussion of $\alpha$ to an earlier part of the manuscript. The vector $\alpha$ measures the alignment of the test point with the eigenspaces of the kernel matrix. This is standard in kernel approximation analyses and introduces no additional modeling assumptions. For smooth kernels (i.e. RBF and Matern) and in-distribution test points, $\alpha$ concentrates in dominant eigendirections, yielding tighter bounds.
>
> **B.3 Discussions added after theorems.**
> We added detailed discussion paragraphs after each main theorem explaining how sketch size, effective dimension, and kernel spectral decay influence approximation quality. Moreover, for predictive mean, variance, and NLML we now provide uniform approximation guarantees over compact domains.
>
> **B.4 Role of $\beta$.**
> Sampling satisfies $p_i\ge \beta \ell_i/\sum_j\ell_j$. For $\beta=1$, exact leverage sampling gives strongest guarantees but costs $\mathcal O(n^3)$. For $\beta<1$, approximate leverage scores are computed in $\mathcal O(nm^2)$ time using Algorithm 2 in paper, appearing in bounds through $d_{\text{eff}}/\beta$. Empirically $\beta\approx0.5$–$0.8$ for $m\approx0.1n$. Smaller $\beta$ requires larger sketch size, made explicit by $m\ge 8(d_{\text{eff}}/\beta)\log(n/\delta)$.
>
> **B.5 Learning rates with respect to $n$.**
>
> We have added new Corollary that explicitly derives learning rate in terms of sample size $n$ under polynomial eigendecay.
>
> Under polynomial eigen-decay $\sigma_i\asymp i^{-\rho}$ with $\rho>1$ and $\sigma_\xi^2\asymp n^{-1}$, assume the test kernel vector satisfies the source condition of order $r>1/2$: $k_* = U\alpha$ and  $\alpha_i^2 \le C \cdot \sigma_i^{1+2r} $,  $i=1,\ldots,n$.
>
> We obtain $d_{\text{eff}}=O(n^{1/\rho})$, hence $m=O(n^{1/\rho}\log n)$ and the predictive mean learning rate
> $ |\mu(x_*) - \mu_S(x_*)| = O(||y||_2 \cdot n^{−(r−\frac{1}{2}−\frac{1}{2ρ})})$.
> This shows how kernel smoothness controls both sketch complexity and learning rate.
>
> ---
>
> ### C) Experiments
>
> **C.1 Synthetic experiments.**
>
> We will include a figure validating theoretical assumptions in the revised manuscript.
>
> **C.2 Clarifying sample size.**
>
> In all our figures, the x-axis labeled "Sample Size in \% of Full Data'' refers to the sketch size $m$ expressed as a percentage of the training set size $n$. For example, "6\%'' means $m \approx 0.06n$ columns are selected from the full kernel matrix.
>
> **C.3 Training time vs evaluation metrics.**
>
> We focus on predictive accuracy (NLPD, MSLL, variance error) since our goal is approximation quality. Training times are **already reported in some tables in appendix**: Nystrom methods, including ours, have similar runtimes, while SVGP is slower due to variational overhead dominates in relatively moderate dataset. As all Nystrom variants share $\mathcal{O}(nm^2)$ complexity, time--accuracy plots are not interesting---the key differentiator is accuracy. While SVGP may scale better for very large dataset via mini-batching due to less memory constraint, its uncertainty quantification is empirically inferior to Nystrom, and such regimes are beyond our scope since exact GP ground truth is infeasible.
>
> While our method has similar runtime to other Nystrom variants, it provides superior predictive performance and uncertainty estimates, making it the preferred choice when high-fidelity GP approximations are required.

---

### Review · Reviewer_Mqyt · 2025-12-03

**Summary Of Contributions:**

Main contribution is establishing error bound for Gaussian process regression when kernel matrix is approximated due computational limitations. Given $(x_i, y_i)$ , the papers estimations an error for estimating $y(x_*)$ where $x_*$ is a single test sample. The main idea is combining approximation error of kernel and using closed form solution of Gaussian processes.
However there are two fundamental issues with the analysis
- (1) the result only holds for a single test sample $(x_*)$ in my opinion all deviation bounds should have sup over $x_*$. Suppose that sketching does well on one single test sample $x_*$. Since there are infinite choice for $x_*$, we can not apply union bound. Thus, the high probability statements in theorems 2 and 3 can not be used in practice when $x_*$ is random.
- (2) High probability statements in theorems 2 and 3 are over randomness of the sketching and again input $x_1, \dots, x_n$ is assumed to be frozen. These bounds are only informative if they holds when $x_1, \dots, x_n$ are random.

**Audience:**

No

**Audience Explanation:**

Please check my comments above

**Broader Impact Concerns:**

N.A.

**Claims And Evidence:**

No

**Claims Explanation:**

In my opinion, machine learning community will not find the established analysis informative as they are mostly numerical approximation errors for a single test sample. Please check my summary for details.

**Requested Changes:**

Minimum correction is proving that the statement of theorem 2 holds for $\sup_{x_*} | \mu(x_*) - \mu_S(x_*)|$ or when $x_1, \dots, x_n, x_*$ are i.i.d. from an unknown distribution. All other theorems require a similar correction.

---

> ### Author Response · Authors · 2026-01-09
> **Clarifying Pointwise vs. Uniform Guarantees and the Role of Sketching Randomness**
>
> We sincerely thank the reviewer for the careful reading and constructive feedback. We clarify both concerns below and summarize the corresponding revisions.
>
> ---
>
> **(1) Pointwise vs. uniform deviation over x\***
>
> We agree that uniform guarantees strengthen the theoretical message. However, we emphasize that the original pointwise bounds are *not vacuous nor impractical*. In Gaussian process regression, predictions are queried at individual test locations sampled from the same environment as the training data. Our original Theorems 2–4 provide high-probability guarantees for the predictive mean, variance, and NLML at any *arbitrary but fixed* test location x\*. Thus, for any random test point encountered at inference time, the bound holds with probability $1 − \delta$ over the sketching randomness, which is the standard interpretation in sketching literature.
>
> However, following the reviewer’s suggestion, we have now added the following uniform results for all our theorem.
>
> **Corollary (Uniform Predictive Mean Approximation).**
> Assume that the kernel is uniformly bounded on a compact domain $\mathcal{X}$, i.e., $\sup_{x \in \mathcal{X}} k(x,x) \le \kappa^2$ and that the sketching parameters satisfy $\sigma_\xi^2 > t\gamma$.
> Then, with probability at least $1-\delta$ over the sketching randomness, the predictive mean error satisfies,
> $$ sup_{x* ∈ X} | μ(x*) − μ_S(x*) | ≤ γκ / ((1 − t) \sqrt{σ^2_ξ}) · ||y||_2 /( σ^2_ξ − tγ ) $$
>
> We also provide analogous uniform corollaries for predictive variance and NLML.
>
> ---
>
> **(2) Randomness over sketching vs. randomness of x₁, …, xₙ**
>
> Our analysis intentionally conditions on the observed dataset and places probability only over the sketching randomness. This follows the standard paradigm in randomized numerical linear algebra and kernel sketching (see, Mahoney [2011], Pilanci and Wainwright [2016], Musco and Musco [2017]), where the data matrix is treated as fixed and randomness is introduced solely through the sketch. This is not a limitation but a design choice: sketching is a *computational approximation mechanism*, not a statistical estimator of the data-generating distribution. Conditioning on the realized dataset ensures that the approximation quality is guaranteed for the exact problem faced by the practitioner.
>
> Thus, our statements are of the form: *for the given dataset (X, y), the sketched GPR solution approximates the exact GPR solution with probability $1 − \delta$ over the sketch*. This is the most practically relevant form of guarantee in scalable GPR.
>
> ---
>
> **Summary of revisions.**
> We clarified the interpretation and practical relevance of the original pointwise bounds, added new uniform sup-norm corollaries, and explicitly state that our probabilistic analysis follows the standard sketching paradigm with fixed data and randomness only in the sketch.
>
> ---
>
> **References**
>
> - Mahoney, M. W. (2011). *Randomized algorithms for matrices and data*. Foundations and Trends in Machine Learning.
> - Pilanci, M., and Wainwright, M. J. (2016). *Randomized sketches of convex programs with sharp guarantees*. Journal of Machine Learning Research (JMLR).
> - Musco, C., and Musco, C. (2017). *Recursive sampling for the Nyström method*. NeurIPS.

---

### Review · Reviewer_gy7Z · 2025-12-30

**Summary Of Contributions:**

Inspired by the ridge leverage score method from El Alaoui & Mahoney (NeurIPS 2015), the submitted paper introduces a sketch-based approximation for Gaussian process regression (GPR). The ridge leverage score method from El Alaoui et al. extends the notion of statistical leverage scores to the setting of kernel ridge regression, so it is possible to identify a sampling distribution that reduces the size of the sketch (required number of columns). In this case, we are developing a similar method for setting low-rank approximations of the kernel matrix of the GPR model. Such a low-rank approximation is used for the estimation of the (posterior) predictive mean, the variance, and the log-marginal likelihood.

**Additional Comments:**

N/A

**Audience:**

Yes

**Audience Explanation:**

I do think low-rank approximations for GPR are always of interest for the TMLR audience. However, I have some concerns regarding how this one is presented in the current submission. In some way, I am afraid that the contributions are not that visible, and the work is just a result of extending the El Alaoui & Mahoney insights for the specific case of predictive means and variance and the LML approximation. In the same way, the story does not really align with well-known approximations or other papers that make advances on the Nystrom method for GPs. Additionally, when considering the inequalities, it's not really discussed in depth what are the dominant terms, or at least, not in a way that make one understand how the number of columns, structure of the sketch of the kernel matrix, or just the approximation are directly affecting the inequality.

**Claims And Evidence:**

No

**Claims Explanation:**

The paper seems to be well-written and clear enough for a reader familiar with GPR models and low-rank approximations. However, the way sketching is presented and how the previous work is revisited is perhaps a bit too quick for my taste, and I had to go back to El Alaoui & Mahoney (NeurIPS 2015) to understand what was going on. In that sense, the paper is not really self-contained in its current state. Additionally, as long as I revisit such previous work, I can see that the results are very close to the ones presented in the current submission, more than what one should expect in this case.

Just to give some particular examples of the previous comments here: i) the Algorithm section 4, with the definition of the ridge leverage score is the same one as in section 3.3 of El Alaoui & Mahoney (NeurIPS 2015), but less clear and adapted for the GPR data-noise parameter. The theoretical guarantees from Section 5 are the same ones introduced in Section 3.1 of El Alaoui & Mahoney, but adapted to mean, variance, and marginal likelihood in the following sections. Even the effective dimension is also taken and adapted.

Having Theorem 3 from El Alaoui & Mahoney (NeurIPS 2015), with the bound on the number of columns for the sketch approximation, and also having the sketch of the kernel matrix $L_{\gamma}$, I see these two results re-used three times for the predictive mean, variance and log-marginal-likelihood approximation. From the perspective of a researcher insterested in GPR, sections 5.2, 5.3 and 5.4 seems a repetition, with the changes only visible on the inequality --- which is not discussed in depth.

**Requested Changes:**

- I would like to see a manuscript which is (at least) as self-contained as El Alaoui & Mahoney (2015).
- The technical results around sections 5.2, 5.3, and 5.4 are somewhat repetitive, and definitions are even repeated. In many ways, one can think that the sketch of the kernel matrix is just being substituted in the original GPR terms. I would like to see this changed in some degree.
- How inequalities are obtained, what are the main differences and contributions (different from the ones El Alaoui & Mahoney) should be much clearer in the manuscript, as well as connecting with the theoretical derivations included in the Appendix, so the reader can follow and reproduce.
- Experimental results are kind of fine, but I am afraid that the metrics provided for the particular UCI datasets don't really show the big picture, as sparse or spectral approximations have other advantages that are omitted in this case.

---

> ### Author Response · Authors · 2026-01-09
> **Novel Theory for Full Bayesian GPR: Predictive Variance and NLML Guarantees Beyond Prior KRR Work**
>
> We sincerely thank the reviewer for the detailed feedback. Below we clarify the scope and novelty of our work and address the main concerns.
>
> ---
>
> ### On reliance on El Alaoui and Mahoney (2015)
>
> Our paper uses ridge leverage score sketching only as a *kernel approximation primitive*, adopting the same sampling distribution and sketch-size bounds as El Alaoui and Mahoney (2015). Beyond this kernel approximation tool, **none of the theoretical results are inherited**.
>
> El Alaoui and Mahoney analyze *kernel ridge regression (KRR)*, which is a point-estimation framework focused on excess prediction risk. In contrast, our work studies *full Bayesian Gaussian Process Regression (GPR)*, whose core quantities are ,
>
> (i) posterior predictive mean,
> (ii) posterior predictive variance, and
> (iii) negative log marginal likelihood (NLML).
>
> Both **predictive variance and NLML do not exist in KRR** and are neither defined nor analyzed in Mahoney’s work. These quantities are fundamental to GPR: predictive variance governs uncertainty quantification, and NLML is required to *learn kernel hyperparameters*, not merely tune them.
>
> In particular, NLML involves the term $\log\det(K+\sigma_\xi^2 I)$, which has no analogue in KRR and cannot be controlled using the bias–variance framework of prior work. Its analysis requires new lower bounds on spectral perturbations of the sketched kernel matrix. Similarly, predictive variance involves quadratic forms of inverse kernel perturbations, which are again outside the scope of KRR theory.
>
> Thus, while ridge leverage sketching is reused as a kernel approximation mechanism, our paper provides the **first theoretical guarantees for predictive variance and NLML under sketching in GPR**, together with predictive mean bounds adapted to the Bayesian setting.
>
> We have additionally derived uniform convergence bounds for all three theorems, as requested by the reviewers, which significantly strengthen the theoretical guarantees of our work.
>
> ---
>
> ### On repetition across theoretical sections
>
> Although the same sketch distortion bound is used as a starting point, the three quantities analyzed (mean, variance, NLML) are mathematically distinct and require different perturbation arguments. This is why separate theorems are necessary and do not reduce to a repetition of KRR analysis.
>
> We will incorporate these clarifications directly into the manuscript and make the exposition fully self-contained, so that all assumptions and definitions are clearly stated without requiring external references.
>
> ---
>
> ### On experimental evaluation
>
> Our experiments focus specifically on predictive mean, predictive variance, NLML, runtime, and memory—exactly the quantities covered by our theory. We further show that ridge leverage score–based Nystrom sketching provides **superior uncertainty quantification** in practice, validating the relevance of our new theoretical guarantees.
>
> We have emphasized this point in the revised manuscript and will ensure that it is clearly conveyed that the primary strength of our approach is its substantially improved uncertainty quantification relative to existing baselines.
>
> ---
>
> ### Summary
>
> Our contribution is not the reuse of ridge leverage sketching itself, but the **first complete theoretical treatment of ridge leverage–based kernel sketching for full Gaussian process inference**, including uncertainty quantification and hyperparameter learning, which are absent from prior KRR-based analyses.

---

### Decision · Action_Editor_xuXF · 2026-02-22

**Recommendation:** Accept with minor revision

**Additional Comments:**

The manuscript must be updated to formally incorporate the clarifications, theoretical proofs, and empirical additions provided during the rebuttal phase.

Specifically, please ensure the following are integrated:

- **Relationship with El Alaoui & Mahoney (2015)**: Clarify and discuss the boundaries of what is borrowed versus what is novel. State clearly that your work only uses ridge leverage score sketching as a kernel approximation primitive and that no theoretical results are inherited beyond this

- **Connections to Random Fourier Features (RFF)**: Incorporate the discussion comparing your approach to RFF into the Related Work section, and include empirical results generated during the rebuttal.

- **Theoretical Clarifications & Additions**:
  - *Deviation Bounds*: Include the discussion on pointwise versus uniform deviation over $x^*$, and add the formal proof of the presented corollary.
  - *Sources of Randomness*: Articulate the distinction between randomness over the sketching process versus randomness of the data points. Please explain why this is the most practically relevant form of guarantee in scalable GPR.
  - *Assumptions*: Update the text to include the clarifications regarding assumptions and other minor technical details discussed with the reviewers.

**Audience:**

Yes

**Audience Explanation:**

The derivation of bounds on the approximation error for the predictive mean, predictive variance, and negative log-likelihood with score-based GPR sketching is a sufficient and interesting contribution to the TMLR audience working on scalable Bayesian methods.

The primary concern raised was that the work may be seen incremental when compared to the foundational paper by El Alaoui & Mahoney (2015).

**Claims And Evidence:**

Yes

**Claims Explanation:**

During the review process, the reviewers found no critical issues with the provided theoretical evidence (see suggestions below), and requested some minor improvements for the experimental evaluation.

---

> ### Author Response · Authors · 2026-03-31
> **Final Note of Thanks**
>
> Dear Action Editor and Reviewers,
>
> Thank you for your time and effort in handling our manuscript and for the constructive feedback throughout the review process. We sincerely appreciate your support and are glad that the paper has been accepted.
>
> Best regards,
>
> Jayesh Malaviya
>
> (on behalf of all authors)